# Regret Minimization Experience Replay in Off-Policy Reinforcement Learning

**Xu-Hui Liu**[*], **Zhenghai Xue**[*], **Jing-Cheng Pang, Shengyi Jiang, Feng Xu, Yang Yu**[†]
National Key Laboratory for Novel Software Technology
Nanjing University, Nanjing 210023, China
liuxh@lamda.nju.edu.cn, xuezh@smail.nju.edu.cn
{pangjc, jiangsy, xufeng}@lamda.nju.edu.cn, yuy@nju.edu.cn

## Abstract

In reinforcement learning, experience replay stores past samples for further reuse. Prioritized sampling is a promising technique to better utilize these samples. Previous criteria of prioritization include TD error, recentness and corrective feedback, which are mostly heuristically designed. In this work, we start from the regret minimization objective, and obtain an optimal prioritization strategy for Bellman update that can directly maximize the return of the policy. The theory suggests that data with higher hindsight TD error, better on-policiness and more accurate Q value should be assigned with higher weights during sampling. Thus most previous criteria only consider this strategy partially. We not only provide theoretical justifications for previous criteria, but also propose two new methods to compute the prioritization weight, namely ReMERN and ReMERT. ReMERN learns an error network, while ReMERT exploits the temporal ordering of states. Both methods outperform previous prioritized sampling algorithms in challenging RL benchmarks, including MuJoCo, Atari and Meta-World.

## 1 Introduction

Reinforcement learning (RL) [1] has achieved great success in sequential decision making problems. Off-policy RL algorithms [2, 3, 4, 5, 6] have the ability to learn from a more general data distribution than on-policy counterparts, and often enjoy better sample efficiency. This is critical when the data collection process is expensive or dangerous. Experience Replay [7] enables data reuse and has been widely used in off-policy reinforcement learning. Previous work [8] points out that emphasizing on important samples in the replay buffer can benefit off-policy RL algorithms. Prioritized Experience Replay (PER) [9] quantifies such importance by the magnitude of temporal-difference (TD) error. Based on PER, many sampling strategies [10, 11, 12] are proposed to perform prioritized sampling. They are either based on TD error [9, 10, 12] or focused on the existence of corrective feedback [11]. However, these are all proxy objectives and different from the objective of RL, i.e., minimizing policy regret. They can be suboptimal in some cases due to this objective mismatch.

In this paper, we first give examples to illustrate the objective mismatch in previous prioritization strategies. Experiments show that lower TD error or more accurate Q function can not guarantee better policy performance. To tackle this issue, we first formulate an optimization problem that directly minimizes the regret of the current policy with respect to prioritization weights. We then make several approximations and solve this optimization problem. An optimal prioritization strategy is obtained and indicates that we should pay more attention to experiences with higher hindsight TD error, better on-policiness and more accurate Q value. To the best of our knowledge, this paper is

---

[*]Equal contribution
[†]Corresponding author

35th Conference on Neural Information Processing Systems (NeurIPS 2021).

the first to optimize the sampling distribution of replay buffer theoretically from the perspective of regret minimization.

We then provide tractable approximations to the theoretical results. The on-policiness can be estimated by training a classifier to distinguish recent transitions, which are generally more on-policy, from early ones, which are generally more off-policy. The oracle Q value is inaccessible during training, so we can not calculate the accuracy of Q value directly. Inspired by DisCor [11], we propose an algorithm named ReMERN which estimates the suboptimality of Q value with an error network updated by Approximate Dynamic Programming (ADP).

ReMERN outperforms previous methods in environments with high randomness, e.g. with stochastic target positions or noisy rewards. However, the training of an extra neural network can be slow and unstable. We propose another estimation of Q accuracy based on a temporal viewpoint. With Bellman updates, the error in Q value accumulates from the next state to the previous one all across the trajectory. The terminal state has no bootstrapping target and low Bellman error. Therefore, states fewer steps away from the terminal state will have lower error in the updated Q value because of the more accurate Bellman target. This intuition is verified both empirically and theoretically. We then propose Temporal Correctness Estimation (TCE) based on the distance of each state to a terminal state, and name the overall algorithm ReMERT.

Similar to PER, ReMERN and ReMERT can be a plug-in module to all off-policy RL algorithms with a replay buffer, including but not limited to DQN [5] and SAC [2]. Experiments show that ReMERN and ReMERT substantially improve the performance of standard off-policy RL methods in various benchmarks.

## 2 Background

### 2.1 Preliminaries

A Markov decision process (MDP) is denoted $(\mathcal{S}, \mathcal{A}, T, r, \gamma, \rho_0)$, where $\mathcal{S}$ is the state space, and $\mathcal{A}$ is the action space. $T(s'|s, a)$ and $r(s, a) \in [0, \mathrm{R}_{\max}]$ are the transition and reward function. $\gamma \in (0, 1)$ is the discounted factor and $\rho_0(s)$ is the distribution of the initial state. The target of reinforcement learning is to find a policy that maximizes the expected return: $\eta(\pi) = \mathbb{E}_\pi[\sum_{t \geq 0} \gamma^t r(s_t, a_t)]$, where the expectation is calculated from trajectories sampled from $s_0 \sim \rho_0$, $a_t \sim \pi(\cdot|s_t)$, and $s_{t+1} \sim T(\cdot|s_t, a_t)$ for $t \geq 0$.

For a fixed policy, an MDP becomes a Markov chain, where the discounted stationary state distribution is defined as $d^\pi(s)$. With a slight abuse of notation, the discounted stationary state-action distribution is defined as $d^\pi(s, a) = d^\pi(s)\pi(a|s)$. Then the expected return can be rewritten as $\eta(\pi) = \frac{1}{1-\gamma}\mathbb{E}_{d^\pi(s,a)}[r(s, a)]$. We assume there exists an optimal policy $\pi^*$ such that $\pi^* = \arg\max_\pi \eta(\pi)$. We use the standard definition of the state-action value function, or Q function: $Q^\pi(s, a) = \mathbb{E}_\pi[\sum_{t \geq 0} \gamma^t r(s_t, a_t)|s_0 = s, a_0 = a]$. Let $Q^*$ be the shorthand for $Q^{\pi^*}$. $Q^*$ satisfies the Bellman equation $Q^*(s, a) = \mathcal{B}^*(Q^*(s, a))$, where $\mathcal{B}^* : \mathbb{R}^{\mathcal{S} \times \mathcal{A}} \to \mathbb{R}^{\mathcal{S} \times \mathcal{A}}$ is the Bellman optimal operator: $(B^* f)(s, a) := r(s, a) + \gamma \max_{a'} \mathbb{E}_{s' \sim P(s,a)} f(s', a')$, where $f \in \mathbb{R}^{\mathcal{S} \times \mathcal{A}}$.

The regret of policy $\pi$ is defined as $\mathrm{Regret}(\pi) = \eta(\pi^*) - \eta(\pi)$. It measures the expected loss in return by following policy $\pi$ instead of the optimal policy. Since $\eta(\pi^*)$ is a constant, minimizing the regret is equivalent to maximizing the expected return, and thus it can be an alternative objective of reinforcement learning.

### 2.2 Related Work

Extensive researches have been conducted on experience replay and replay buffer. The most frequently considered aspect is the sampling strategy. Various techniques have achieved good performance by performing prioritized sampling on the replay buffer. In model-based planning, Prioritized Sweeping [13, 14, 15] selects the next state updates according to changes in value. Prioritized Experience Replay (PER) [9] prioritizes samples with high TD error. Taking PER one step further, Prioritized Sequence Experience Replay (PSER) [10] considers information provided by transitions when estimating TD error. Emphasizing Recent Experience (ERE) [16] and Likelihood-Free Importance Weighting (LFIW) [12] prioritizes the correction of TD errors for frequently encountered

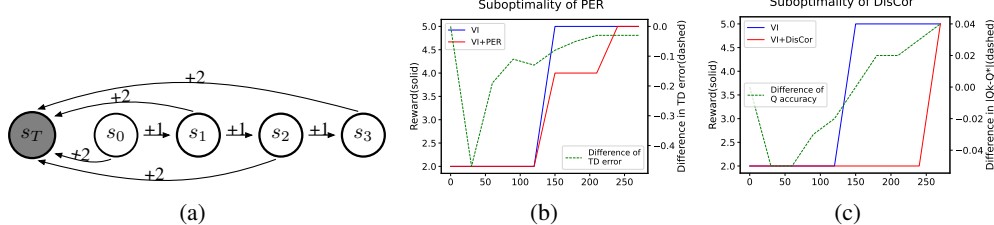

(a)                  (b)                  (c)

Figure 1: A simple MDP showing the objectives of PER and DisCor can slow down the training process. (a) A 5-state MDP with initial state $s_0$ and terminal state $s_T$. Except for $s_3$ and $s_T$, there are two available actions, *left* and *right*. Turning *left* leads to the terminal state $s_T$ and +2 reward, while turning *right* leads to the next state and +1 reward. The optimal policy is to keep turning *right* until reaching $s_3$, then reach $s_T$. (b) Relationship between TD error (dashed line) and performance (solid line) of VI and VI+PER. (c) Relationship between Q error (dashed line) and performance (solid line) of VI and VI+DisCor.

states. Distribution Correction (DisCor) [11] assigns higher weights to samples with more accurate target Q value because these samples provide "corrective feedback". DisCor uses a neural network to estimate the accuracy of target Q value. Inspired by DisCor, SUNRISE [17] proposes to use the variance of ensembled Q functions as a surrogate for the accuracy of Q value. Adversarial Feature Matching [18] focuses on sampling uniformly among state-action pairs.

Instead of proposing a new strategy of sampling, [19] proves that there exists a relationship between sampling strategy and loss function, and weighted value loss can serve as a surrogate for prioritized sampling. Other works focus on buffer capacity [20, 21]. They point out that a proper buffer capacity can accelerate value estimation and lead to better learning efficiency and performance. In fact, this can be thought of as a specific example of prioritization strategies, i.e., assigning zero weights to the samples exceeding the proper buffer capacity.

## 3 Optimal Prioritization Strategy via Regret Minimization

### 3.1 Revisiting Existing Prioritization Methods

PER and DisCor are two representative algorithms of prioritized sampling. PER prioritizes state-action pairs with high TD error, while DisCor prefers to perform Bellman update on state-action pairs that have more accurate Bellman targets. However, both criteria are different from the target of RL algorithms, which is to maximize the expected return of the policy. Such difference can slow down the training process in some cases. For example, when the Bellman target is inaccurate, minimizing TD error does not necessarily improve the optimality of Q value.

To illustrate the aforementioned problems, we provide an example MDP shown in the left part of Fig. 1(a). This is a five-state MDP with two actions: turning left and right. The optimal policy is to turn *right* in all states, receiving a total reward of 5. Suppose the Q values for all $(s, a)$ pairs are initialized to zero. The reward of turning *left* is higher than turning *right* in all states, so the *left* action has a higher TD error. As a result, PER prefers states with the *left* action, which is not the fastest training process to achieve the optimal policy. Also, since there is no bootstrapping error for the terminal state $s_T$, transitions with $s_T$ as the next state have an accurate target Q value. Therefore, DisCor also focuses on state-actions pairs with *left* action, which is again not optimal.

We perform Value Iteration (VI) on this MDP. To simulate function approximation in Deep RL and avoid convergence in few iterations, the learning rate is set to $0.1$. Prioritized sampling is substituted by weighted Bellman update, as introduced in [19]. The results are shown in Fig. 1(b) and Fig. 1(c). According to the results, PER indeed minimizes TD error more efficiently, and DisCor results in a more accurate estimation of Q value, as indicated by their objectives. However, they both need more iterations to converge than value iteration without prioritization. According to this MDP, the objective of previous prioritization methods can be inefficient in certain cases.

## 3.2 Problem Formulation of Regret Minimization

As shown in Section 3.1, an indirect objective can cause slower convergence of value iteration. In this section, we aim to find an optimal prioritization weight $w_k$ that can directly minimize the policy regret $\eta(\pi^*) - \eta(\pi_k)$. The weight is multiplied to the Bellman error $(Q - \mathcal{B}^*Q_{k-1})^2$ and $\pi_k$ can be obtained from the updated Q function. To facilitate further derivations, we only consider the best Q function of the Bellman update, which is calculated by the $\arg\min$ operator. Therefore, the optimization problem with respect to $w_k$ can be written as:

$$
\min_{w_k} \quad \eta(\pi^*) - \eta(\pi_k)
$$
$$
\text{s.t.} \quad Q_k = \arg\min_{Q \in \mathcal{Q}} \; \mathbb{E}_\mu[w_k(s,a) \cdot (Q - \mathcal{B}^*Q_{k-1})^2(s,a)], \tag{1}
$$
$$
\mathbb{E}_\mu[w_k(s,a)] = 1, \quad w_k(s,a) \geq 0,
$$

where $\pi_k(s) = \frac{\exp(Q_k(s,a))}{\sum_{a'}\exp(Q_k(s,a'))}$ is the policy corresponding to $Q_k$. $\mathcal{Q}$ is the function space of Q functions and $\mu$ is the data distribution of the replay buffer. $Q_k$ is the estimate of Q value after the Bellman update at iteration $k$.

We then manage to solve this optimization problem. To get started, we introduce *recurring probability* which serves as an upper bound of the error term in our solution.

**Definition 1** (Recurring Probability). *The recurring probability of a policy $\pi$ is defined as $\epsilon_\pi = \sup_{s,a} \sum_{t=1}^\infty \gamma^t \rho^\pi(s,a,t)$, where $\rho$ is the probability of the agent starting from $(s,a)$ and coming back to $s$ at time step $t$ under policy $\pi$, i.e., $\rho^\pi(s,a,t) = \Pr(s_0 = s, a_0 = a, s_t = s, s_{1:t-1} \neq s; \pi)$.*

We then present the solution to the optimization problem 1 in Thm. 1. The formal version of the theorem and detailed proof are in Appendix A.

**Theorem 1** (Informal). *Under mild conditions, the solution $w_k$ to a relaxation of the optimization problem 1 in MDPs with discrete action spaces is*

$$
w_k(s,a) = \frac{1}{Z_1^*}\left(E_k(s,a) + \epsilon_{k,1}(s,a)\right). \tag{2}
$$

*In MDPs with continuous action spaces, the solution is*

$$
w_k(s,a) = \frac{1}{Z_2^*}\left(F_k(s,a) + \epsilon_{k,2}(s,a)\right). \tag{3}
$$

*where*

$$
E_k(s,a) = \underbrace{\frac{d^{\pi_k}(s,a)}{\mu(s,a)}}_{(a)} \underbrace{(2 - \pi_k(a|s))}_{(b)} \underbrace{\exp\left(-|Q_k - Q^*|(s,a)\right)}_{(c)} \underbrace{|Q_k - \mathcal{B}^*Q_{k-1}|(s,a)}_{(d)}
$$

$$
F_k(s,a) = 2 \underbrace{\frac{d^{\pi_k}(s,a)}{\mu(s,a)}}_{(a)} \underbrace{\exp\left(-|Q_k - Q^*|(s,a)\right)}_{(c)} \underbrace{|Q_k - \mathcal{B}^*Q_{k-1}|(s,a)}_{(d)},
$$

$Z_1^*$, $Z_2^*$ *are normalization factors,* $\epsilon_{k,1}(s,a)$ *and* $\epsilon_{k,2}(s,a)$ *satisfy* $\max\left\{\frac{\epsilon_{k,1}(s,a)}{E_k(s,a)}, \frac{\epsilon_{k,2}(s,a)}{F_k(s,a)}\right\} \leq \epsilon_{\pi_k}$.

With regard to the error terms, there are two cases where $\epsilon_{\pi_k}$ is low by its definition: the probability of coming back to the states that have been visited is small, or the number of steps an agent takes to come back to the visited states is large. In most tasks, either of these cases holds. We conduct experiments in several Atari games and show the verification results in Appendix D. The low probability leads to small $\epsilon_{\pi_k}$ and implies the terms $\epsilon_{k,1}(s,a)$ and $\epsilon_{k,2}(s,a)$ are negligible.

Therefore, Thm. 1 suggests that state-action tuples in the replay buffer should be assigned with higher importance if they have the following properties:

- **Higher hindsight Bellman error** ( from $|Q_k - \mathcal{B}^*Q_{k-1}|(s,a)$). $Q_k$ is the estimate of Q value after the Bellman update. This term describes the difference between the estimated hindsight Q value and the Bellman target. It is similar to the prioritization criterion of PER [9], but PER concerns more about the historical Bellman error, i.e., $|Q_{k-1} - \mathcal{B}^*Q_{k-2}|(s,a)$.

- **More on-policiness** ( from $\frac{d^{\pi_k}(s,a)}{\mu(s,a)}$ ). An efficient update of $\pi$ requires $w_k$ to be on-policy, i.e., focusing on state-action pairs which are more likely to be visited by the current policy. Such prioritization strategy has been empirically illustrated in LFIW [12] and BCQ [22], while we obtain it directly from our theorem.

- **Closer value estimation to oracle** ( from $\exp\left(-\left|Q_k - Q^*\right|(s,a)\right)$ ). This term indicates that state-action pairs with less accurate Q values after the Bellman update should be assigned with lower weights. Intuitively, state-action pairs that lead to suboptimal updates of the estimator of Q value should be down-weighted. Such suboptimality may arise from incorrect target Q values or the error of function approximation in deep Q networks.

- **Smaller action likelihood** (from $2 - \pi_k(a|s)$). This term only exists in MDPs with a discrete action space. It offsets the effect of the on-policy term $d^{\pi_k}$ to some extent and is similar to $\varepsilon$-greedy strategy in exploration.

Our theoretical analysis indicates that existing prioritization strategies only consider the problem partially, neglecting other terms in minimizing the regret. For example, DisCor fails to consider the on-policiness and PER ignores the accuracy of Q value. In the remaining part of this section, we present practical approximations to each term in Eq. (2) and (3).

Term (a) is the importance weight between the current policy and the behavior policy. We can calculate this term using Likelihood-Free Importance Weighting (LFIW, [12]). LFIW divides the replay buffer into two parts, a fast buffer $\mathcal{D}_f$ and a slow buffer $\mathcal{D}_s$. It initializes a neural network $\kappa_\psi(s,a)$ and optimizes the network according to the following loss function:

$$L_\kappa(\psi) := \mathbb{E}_{\mathcal{D}_s}\left[f^*\left(f'\left(\kappa_\psi(s,a)\right)\right)\right] - \mathbb{E}_{\mathcal{D}_f}\left[f'\left(\kappa_\psi(s,a)\right)\right], \tag{4}$$

where $f'$ and $f^*$ is the derivative and convex conjugate of function $f$. The updated $\kappa_\psi$ is the desired importance weight.

For term (b) and (d), since $\pi_k$ and $Q_k$ are the policy and the estimate of Q value after the update, they cannot be calculated directly. Therefore, we approximate them by the upper and lower bounds. For term (b), $1 \leq 2 - \pi_k(a|s) \leq 2$. For term (d), a viable approximation is to bound it between the minimum and maximum Bellman errors obtained at the previous iteration, $c_1 = \min_{s,a}|Q_{k-1} - \mathcal{B}^*Q_{k-2}|$ and $c_2 = \max_{s,a}|Q_{k-1} - \mathcal{B}^*Q_{k-2}|$. As shown in DisCor, we can restrict the support of state-action pairs $(s,a)$ used to compute $c_1$ and $c_2$ in the support of replay buffer, to ensure that both $c_1$ and $c_2$ are finite. With these approximation, we can derive a lower bound for $w_k$, which will be detailed in Sec. 3.3 and Sec. 3.4.

In the next two subsections, we will provide two practical algorithms to estimate $|Q_k - Q^*|$.

## 3.3 Regret Minimization Experience Replay Using Neural Network (ReMERN)

DisCor shows $\Delta_k$ can be a surrogate of $|Q_k - Q^*|$, which is defined as:

$$\Delta_k = \sum_{i=1}^k \gamma^{k-i} \left(\prod_{j=i}^{k-1} P^{\pi_j}\right) |Q_i - \mathcal{B}^*Q_{i-1}| \tag{5}$$

$$\implies \Delta_k = |Q_k - \mathcal{B}^*Q_{k-1}| + \gamma P^{\pi_{k-1}}\Delta_{k-1} \tag{6}$$

According to Eq. (6), $\gamma[P^{\pi_{k-1}}\Delta_{k-1}](s,a) + c_2$ is an upper bound of $|Q_k - Q^*|$. This is because $\Delta_k$ is proven to be the upper bound of $|Q_k - Q^*|$ [11] and $c_2$ is the upper bound of $|Q_k - \mathcal{B}^*Q_{k-1}|$. Recall that $2 - \pi_k(a|s) \geq 1$ and $|Q_{k-1} - \mathcal{B}^*Q_{k-2}| \geq c_1$, and we can derive the final expression for this tractable approximation for $w_k(s,a)$ by simplifying all constants:

$$w_k(s,a) \propto \frac{d^{\pi_k}(s,a)}{\mu(s,a)} \exp\left(-\gamma\left[P^{\pi_{k-1}}\Delta_{k-1}\right](s,a)\right), \tag{7}$$

This approximation applies to MDPs with discrete action space and MDPs with continuous action space. Using the lower bound of $w_k(s,a)$ may down-weight some transitions, but will never up-weight a transition by mistake [11].

We use a neural network to estimate $\Delta_{k-1}$. As shown in Eq. (6), $\Delta_{k-1}$ can be calculated from a bootstrapped target, which inspires us to use ADP algorithms to update it. We name this method

ReMERN (**Re**gret **M**inimization **E**xperience **R**eplay using **N**eural Network). The pseudo code for ReMERN is presented in Appendix C. ReMERN is applicable to all value-based off-policy algorithms with replay buffer.

### 3.4 Regret Minimization Experience Replay Using Temporal Structure (ReMERT)

ReMERN uses neural network as the estimator of $|Q_k - Q^*|$. However, training a neural network is time consuming and suffers from large estimation error without adequate iterations. To mitigate this issue, we propose another estimation of $|Q_k - Q^*|$ from a different perspective.

#### 3.4.1 The Temporal Property of Q Error

$|Q_k - Q^*|$ can be decomposed with the triangle inequality: $|Q_k - Q^*| \leq |Q_k - \mathcal{B}^* Q_{k-1}| + |\mathcal{B}^* Q_{k-1} - Q^*|$. The first term is the projection error depending on the Q function space $\mathcal{Q}$. This error is usually small thanks to the strong expressive power of neural networks. In the second term, $\mathcal{B}^* Q_{k-1}$ is the estimate of target Q value, and $|\mathcal{B}^* Q_{k-1} - Q^*|$ is the distance from the target Q value to the ground-truth Q value. The target Q value at the terminal state consists of the reward only, so there is no bootstrapping error and $|\mathcal{B}^* Q_{k-1} - Q^*| = 0$. Moving backward through the trajectory, the accuracy of the Q value estimation decreases as the error of Bellman update accumulates. These Q values are then utilized to compute the target Q value, leading to more erroneous Bellman updates and larger $|\mathcal{B}^* Q_{k-1} - Q^*|$. Such error can accumulate through the MDP. Consequently, states closer to the terminal state tend to have a more accurate Bellman target. This motivates us to estimate the incorrectness of the estimated Q value using the temporal information of a given state-action tuple $(s_t, a_t)$.

To verify our intuition on the temporal property of Q error, we use a gridworld MDP from [23] and visualize the mean error of the target Q value (i.e., $|\mathcal{B}^* Q_{k-1} - Q^*|$) across different actions in Fig. 2. We use DQN to update Q values. In this gridworld MDP, an agent starts at the red triangle on the top-left and terminates at the green rectangle on the top-right. The agent can't go through the wall, which is plotted as gray grids. The darker a grid is, the higher error of Q function it has. This figure illustrates that states closer to the terminal state has lower Q error, corresponding to our intuition that $|\mathcal{B}^* Q_{k-1} - Q^*|$ is related to the position of $(s, a)$ in the trajectory.

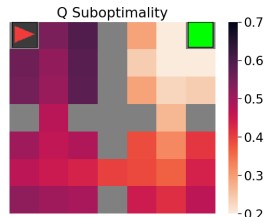

Figure 2: The visualized error of target Q value in a GridWorld Environment. The Q error is visualized by the color of the grid.

To formalize this intuition, we first define *Distance to End*.

**Definition 2** (Distance to End). *Given a MDP $\mathcal{M}$, $\tau = \{s_t, a_t\}_{t=0}^T$ is a trajectory generated by policy $\pi$ in $\mathcal{M}$. The distance to end of $(s_t, a_t)$, denoted by $h_\tau^\pi(s_t, a_t)$, is $T - t$ in this trajectory.*

Our intuition states that the value of $|Q_k - Q^*|$ has a positive correlation with distance to end. Based on this intuition, we propose the following theorem.

**Theorem 2** (Informal). *Under mild conditions, with probability at least $1 - \delta$, we have*

$$
|Q_k(s, a) - Q^*(s, a)|
$$
$$
\leq \mathbb{E}_\tau \left( f(h_\tau^{\pi_k}(s, a))(L_{Q_{k-1}} + c) + \gamma^{h_\tau^{\pi_k}(s,a)+1} c \right) + g(k, \delta) \tag{8}
$$

*where $c = \max_{s,a} \left( Q^*(s, a^*) - Q^*(s, a) \right)$, $f(t) = \frac{\gamma - \gamma^t}{1 - \gamma}$, $L_{Q_{k-1}} = \mathbb{E}[|Q_{k-1} - \mathcal{B}^* Q_{k-2}|]$ and $g(k, \delta)$ decreases exponentially as $k$ increases.*

The formal version of the theorem and its proof are in Appendix B. The theorem states that $|Q_k - Q^*|$ is upper bounded by a function of distance to end and expected Bellman error with high probability.

#### 3.4.2 A Practical Implementation

In Thm. 2 we derive the upper bound of $|Q_k - Q^*|$, which can serve as a surrogate to $|Q_k - Q^*|$. Using an upper bound as the surrogate may down-weight some transitions, but will never up-weight

a transition that should not be up-weighted [11]. We call this Temporal Correctness Estimation (TCE):

$$
\begin{aligned}
|Q_k(s,a) - Q^*(s,a)| &\approx \mathbb{E}_\tau \mathrm{TCE}_c(s,a) \\
&= \mathbb{E}_\tau \left( f(h_\tau^{\pi_{k-1}}(s,a))\big(L_{Q_{k-1}} + c\big) + \gamma^{h_\tau^{\pi_{k-1}}(s,a)+1} c \right),
\end{aligned}
\tag{9}
$$

Similar to the derivation of ReMERN, we can simplify the expression of $w_k(s,a)$ as:

$$
w_k(s,a) \propto \frac{d^{\pi_k}(s,a)}{\mu(s,a)} \exp\left( -\mathbb{E}_\tau \mathrm{TCE}_c(s,a) \right)
\tag{10}
$$

This approach of computing prioritization weights is named ReMERT (**Re**gret **M**inimization **E**xperience **R**eplay using **T**emporal Structure). Its pseudo code is presented in Appendix C. In practice, we record the *distance to end* of a state-action pair when it is sampled by the policy and stored in the replay buffer. The expectation with respect to $\tau$ is computed based on the record and Monte-Carlo estimation.

### 3.5  Comparison between ReMERN and ReMERT

ReMERT can estimate $|Q_k - Q^*|$ directly from the temporal ordering of states, which often provides more efficient and more accurate estimation than ReMERN. However, The expectation with respect to trajectory $\tau$ in Eq. (10) induces statistical error. In some environments, the *distance to end* of a certain state-action pair $(s,a)$ can vary widely across different trajectories, which is usually caused by the randomness of environments. For example, in environments with stochastic goal positions, the state may be near the goal in one episode but far away from it in another. In such cases, prioritization weights provided by ReMERT have large variance and can be misleading. In contrast, ReMERN need to train an error net but is irrelevant to the *distance to end*. Therefore, ReMERN suffers estimation error of neural network but is robust to the randomness of environments. We test their property in the following section.

## 4  Experiments

In this section, we conduct experiments to evaluate ReMERN and ReMERT[3]. We choose SAC and DQN as the baseline algorithms for continuous and discrete action space respectively and incorporate ReMERN and ReMERT as the sampling strategy. We first compare the performance of ReMERN and ReMERT to prior sampling methods in continuous control benchmarks including Meta-World [24], MuJoCo [25] and Deepmind Control Suite (DMC) [26]. We also evaluate our methods in Arcade Learning Environments with discrete action spaces. Then, we dive into our algorithms and design several experiments, such as Gridworld tasks and MuJoCo with reward noise, to demonstrate some key properties of ReMERN and ReMERT. A detailed description of the environments and experimental details are listed in Appendix D.

### 4.1  Performance on Continuous Control Environments

In MuJoCo and DMC tasks, ReMERT outperforms baseline methods on four of six tasks and achieves comparable performance in the rest two tasks, i.e. HalfCheetah and Hopper, as shown in Fig. 3. The marginal improvement of ReMERT in HalfCheetah mainly comes from the absence of a strong correlation between Q-loss and time step. In HalfCheetah, there is no specific terminal state, so the agent always reaches the max length of the trajectory, which gives a fake "distance to end" for every state. In Hopper, there is not much difference of the $|Q_k - Q^*|$ term between all the sampled state-action pairs, as shown in Appendix D, so the state-action pairs are not sampled very unequally. Besides, Hopper is a relatively easy task, in which prioritizing the samples have minor impact on the overall performance of the RL algorithm. The performance of ReMERN is better than DisCor, but is not as good as ReMERT. This verifies our theory and the existence of large estimation error induced by updating neural network with ADP algorithms.

---

[3]Codes are available at https://github.com/AIDefender/ReMERN-ReMERT.

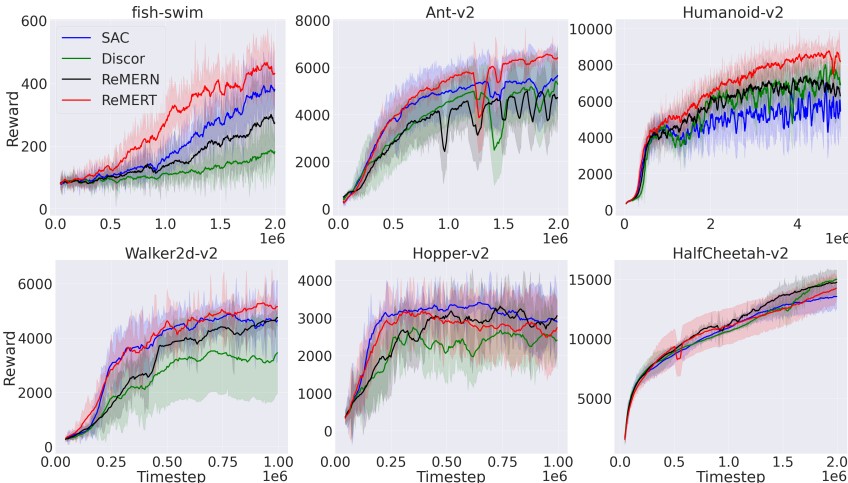

Figure 3: Performance of ReMERT, ReMERN with SAC and DisCor as baselines on continuous control tasks.

The Meta-World benchmark [24] includes many robotic manipulation tasks. We select 8 tasks for evaluation, and plot the result in Fig. 4. The performance of PER can be found in its paper [9]. Current state-of-the-art off policy RL algorithms such as SAC performs poorly on this benchmark because the goals of tasks have high randomness. Although DisCor [11] shows preferable performance in these tasks compared to SAC and PER, ReMERN obtains a significant improvement over DisCor in the training speed and asymptotic performance. In this evaluation, we exclude ReMERT for comparison because the randomized target position in Meta-World contradicts its assumption.

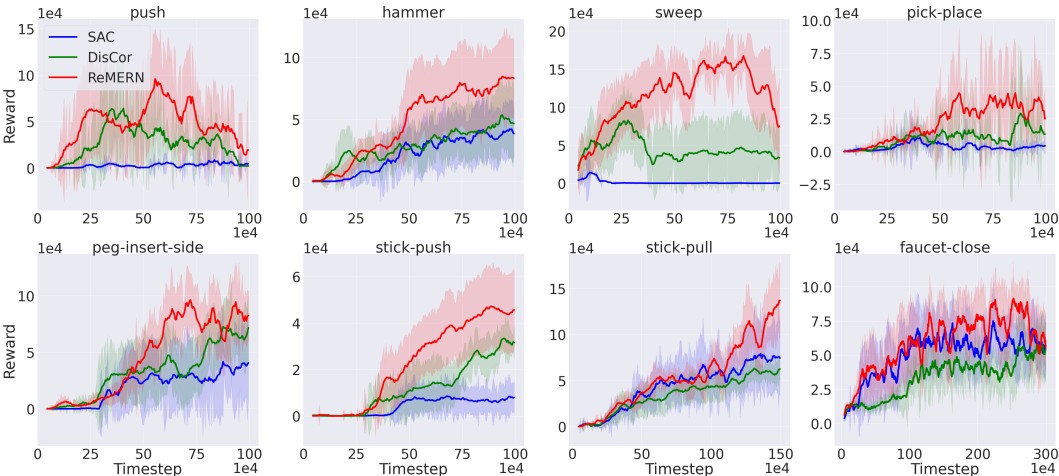

Figure 4: Performance of ReMERN, standard SAC and DisCor in eight Meta-World tasks. From left to right: push, hammer, sweep, peg-insert-side, stick push, stick pull, faucet close.

## 4.2 Performance on Arcade Learning Environments

Atari games are suitable for verifying our theory for MDPs with discrete action space. The tested games have a relatively stable temporal ordering of states because the initial state and the terminal state have little randomness, so that the assumption of ReMERT is satisfied. As shown by Tab. 1, ReMERT outperforms DQN in all the selected games. The results also suggest that ReMERT can be applied to environments with high dimensional state spaces. Results of more Atari games are listed in Appendix D. We do not include ReMERN for comparison because DisCor which is a composing part of ReMERN has no open-source code available for discrete action space.

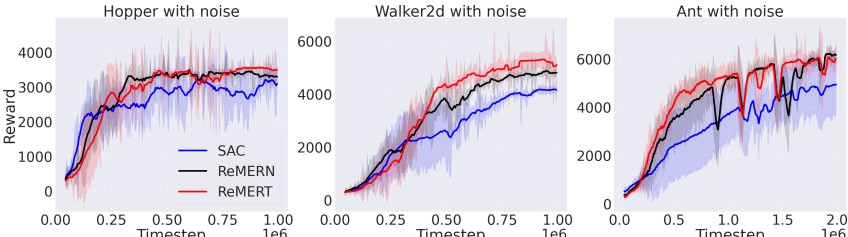

Figure 5: Performance of ReMERN, ReMERT and SAC on three continuous control tasks with reward noise.

### 4.3 Demonstration on Key Properties of ReMERN and ReMERT

#### 4.3.1 Influence of Environment Randomness

Fig. 3 and Fig. 4 show that ReMERN has a better performance on Meta-World than on Mujoco tasks. We attribute this to the robustness of our strategy in environments with high randomness. For a highly stochastic environment, the estimation of Q value is difficult. When the estimation of Q value is inaccurate, the target Q value is also inaccurate, leading to a suboptimal update process in off-policy RL algorithms. Thanks to the closer value estimation to oracle principle, ReMERN estimates the Q value more accurately than other methods. However, for less stochastic environments like MuJoCo environments, the accuracy of error network might become the bottleneck of ReMERN.

To show this empirically, we add Gaussian noise to the reward function in MuJoCo environments. The details of the experimental setup are listed in Appendix D. Fig. 5 show that: (1) ReMERN and ReMERT perform better than SAC in stochastic environments, which verifies our analysis. (2) Though ReMERT suffers statistical error of temporal ordering, it is robust to the randomness of reward because the temporal property is not affected by the noise.

#### 4.3.2 Analysis of TCE on Deterministic Tabular Environments

To analyze the effect of the principle behind TCE, we evaluate the Q error in Gridworld with image input. We plot the $|Q_k - Q^*|$ error of standard DQN, DQN with DisCor, DQN with TCE and DQN with oracle at some time in the training process in Fig. 6. TCE is combined with DQN to estimate term (c) in Eq. (2) , and the other terms are ignored to compute $w_k$. DQN with oracle uses the ground-truth error $|Q_k - Q^*|$ to calculate the prioritization weight. The result shows that DQN with TCE achieves a more accurate Q value estimator than those of standard DQN and DQN with DisCor, while DQN with oracle $|Q_k - Q^*|$ achieves the most accurate Q value estimator. The lower efficiency of DQN with DisCor is due to the slower convergence speed of the error network. This proves the principle behind our theory effective, and TCE is a decent approximation of $|Q_k - Q^*|$.

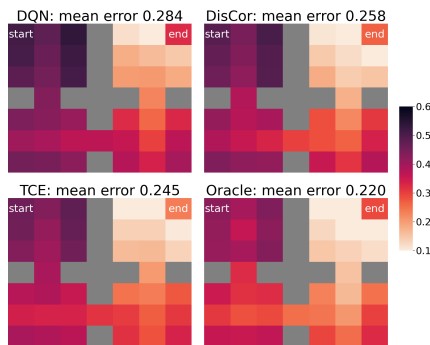

Figure 6: TCE and DisCor in Gridworld

Table 1: DQN vs ReMERT on Atari. DQN (Nature) is the performance in the DQN paper [5]. DQN (Baseline) is the performance of our baseline program [27].

| Method | Enduro | KungFuMaster | Kangaroo | MsPacman | Qbert |
|---|---|---|---|---|---|
| DQN (Nature) | 301±24.6 | 23270±5955 | 6740±2959 | 2311±525 | 10596±3294 |
| DQN (Baseline) | 1185±100 | 29147±7280 | 6210±1007 | 3318±647 | 13437±2537 |
| ReMERT (Ours) | **1303**±258 | **35544**±8432 | **7572**±1794 | **3481**±1351 | **14511**±1138 |

# 5 Conclusion and Future Work

In this work, we first revisit the existing methods of prioritized sampling and point out that the objectives of these methods are different from the objective of RL, which can lead to a suboptimal training process. To solve this issue, we analyze the prioritization strategy from the perspective of regret minimization, which is equivalent to return maximization in RL. Our analysis gives a theoretical explanation for some prioritization methods, including PER, LFIW and DisCor. Based on our theoretical analysis, we propose two practical prioritization strategies, ReMERN and ReMERT, that directly aims to improve the policy. ReMERN is robust to the randomness of environments, while ReMERT is more computational efficient and more accurate in environments with a stable temporal ordering of states. Our approaches obtain superior results compared to previous prioritized sampling methods. Future work can be conducted in the following two directions. First, the framework to obtain the optimal distribution in off-policy RL can be generalized to model-based RL and offline RL. Second, the two proposed algorithms are suitable for different kinds of MDP, so finding a unified prioritization method for all MDPs can further improve the performance.

## Acknowledgements and Disclosure of Funding

We thank Xintong Qi and Xiaolong Yin for helpful discussions. We would also like to thank two groups of anonymous reviewers for their valuable comments on our paper. This work is supported by National Key Research and Development Program of China (2020AAA0107200) and NSFC(61876077).

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
