# A Proof of Theorem 1

In this section, we present detailed proofs for the theoretical derivation of Thm. 1, which aims to solve the following optimization problem:

$$
\begin{aligned}
\min_{w_k} \quad & \eta(\pi^*) - \eta(\pi_k) \\
\text{s.t.} \quad & Q_k = \arg\min_{Q \in \mathcal{Q}} \mathbb{E}_\mu[w_k(s,a) \cdot (Q - \mathcal{B}^* Q_{k-1})^2(s,a)], \\
& \mathbb{E}_\mu[w_k(s,a)] = 1, \quad w_k(s,a) \geq 0,
\end{aligned}
\tag{1}
$$

The problem is equivalent to:

$$
\begin{aligned}
\min_{p_k} \quad & \eta(\pi^*) - \eta(\pi_k) \\
\text{s.t.} \quad & Q_k = \arg\min_{Q \in \mathcal{Q}} \mathbb{E}_{p_k}[(Q - \mathcal{B}^\pi Q_{k-1})^2(s,a)] \\
& \sum_{s,a} p_k(s,a) = 1, \quad p_k(s,a) \geq 0,
\end{aligned}
\tag{2}
$$

The desired $w_k(s,a)$ is $\frac{p_k(s,a)}{\mu(s,a)}$, where $p_k(s,a)$ is the solution to the problem 2.

To solve Problem 2, we need to give the definition of *total variation distance*, *Wasserstein metric* and the diameter of a set, and introduce some mild assumptions.

**Definition 1** (total variation distance)**.** *The total variation (TV) distance of distribution $P$ and $Q$ is defined as*

$$
D_{\text{TV}}(P,Q) = \frac{1}{2} \|P - Q\|_1
$$

**Definition 2** (Wasserstein metric)**.** *For $F, G$ two c.d.fs over the reals, the Wasserstein metric is defined as*

$$
d_p(F,G) := \inf_{U,V} \|U - V\|_p
$$

*where the infimum is taken over all pairs of random variables $(U,V)$ with respective cumulative distributions $F$ and $G$.*

**Definition 3.** *The diameter of a set $A$ is defined as*

$$
\text{diam}(A) = \sup_{x,y \in A} m(x,y)
$$

*where $m$ is the metric on $A$.*

**Assumption 1.** *The state space $\mathcal{S}$ and action space $\mathcal{A}$ are metric spaces with a metric $m$.*

**Assumption 2.** *The $Q$ function is continuous with respect to $\mathcal{S} \times \mathcal{A}$.*

**Assumption 3.** *The transition function $T$ is continuous with respect to $\mathcal{S} \times \mathcal{A}$ in the sense of Wasserstein metric, i.e.,*

$$
\lim_{(s,a) \to (s_0,a_0)} d_p(T(\cdot|s,a), T(\cdot|s_0,a_0)) = 0,
$$

*where $d_p$ denote the Wasserstein metric.*

These assumptions are not strong and can be satisfied in most of environments includes MuJoCo, Atari games and so on.

Let $d_i^\pi(s)$ denote the discounted state distribution, where the state is visited by the agent for the i-th time. that is

$$
d_i^\pi(s) = (1-\gamma) \sum_{t_i=0}^{\infty} \gamma^{t_i} \Pr(s_{t_k} = s, \forall k \in [i]),
$$

where $[k] = \{j \in \mathbb{N}_+ : j \leq k\}$. Notably,

$$d^\pi(s) = \sum_{i=1}^{\infty} d_i^\pi(s) \tag{3}$$

$$d_i^\pi(s) = \sum_{t=1}^{\infty} \rho^\pi(s, \pi(s), t) \gamma^t d_{i-1}^\pi(s), \tag{4}$$

where $\rho^\pi(s, \pi(s), t)$ is the shorthand for $\mathbb{E}_{a \sim \pi} \rho^\pi(s, a, t)$.

The standard definitions of Q function, value function and advantage function is:

$$Q^\pi(s, a) = \mathbb{E}_\pi[\sum_{t \geq 0} \gamma^t r(s_t, a_t) | s_0 = s, a_0 = a].$$

$$V^\pi(s) = \mathbb{E}_\pi[\sum_{t \geq 0} \gamma^t r(s_t, a_t) | s_0 = s].$$

$$A^\pi(s, a) = Q^\pi(s, a) - V^\pi(s).$$

In the follows, Lemma 1 is a technique used in Lemma 2. Lemma 2 shows that $\left|\frac{\partial d^\pi(s)}{\partial \pi(s)}\right|$ is a small quantity.

**Lemma 1.** *Let $f$ be an Lebesgue integrable function, $P$ and $Q$ are two probability distributions, $|f| \leq C$, then*

$$\left| \mathbb{E}_{P(x)} f(x) - \mathbb{E}_{Q(x)} f(x) \right| \leq C D_{\text{TV}}(P, Q) \tag{5}$$

*Proof.*

$$
\begin{aligned}
\left| \mathbb{E}_{P(x)} f(x) - \mathbb{E}_{Q(x)} f(x) \right| &= \left| \sum_x [P(x)f(x) - Q(x)f(x)] \right| \\
&= \left| \sum_x [P(x)f(x) - Q(x)f(x)]\mathbb{I}[P(x) > Q(x)] \right. \\
&\quad \left. - \sum_x [P(x)f(x) - Q(x)f(x)]\mathbb{I}[P(x) < Q(x)] \right| \\
&\leq C D_{\text{TV}}(P, Q)
\end{aligned}
$$

$\square$

**Lemma 2.** *Let $\epsilon_\pi = \sup_{s,a} \sum_{t=1}^{\infty} \gamma^t \rho^\pi(s, a, t)$, we have*

$$\left| \frac{\partial d^\pi(s)}{\partial \pi(s)} \right| \leq \epsilon_\pi d_1^\pi(s) \tag{6}$$

*and $\epsilon_\pi \leq 1$.*

*Proof.* The definition of $\rho^\pi(s, a, t)$ implies

$$0 \leq \sum_{t=1}^{\infty} \gamma^t \rho^\pi(s, a, t) \leq \epsilon_\pi \leq 1, \qquad \forall a \in \mathcal{A}$$

Based on this fact, we have

$$\left| \sum_{t=1}^{\infty} \gamma^t \left( \rho^\pi(s, a_1, t) - \rho^\pi(s, a_2, t) \right) \right| \leq \epsilon_\pi, \qquad \forall a_1, a_2 \in \mathcal{A}$$

Let $\rho^\pi(s, \pi(s), t)$ be a shorthand for $\mathbb{E}_{a \sim \pi(s)} \rho^\pi(s, a, t)$.

If $\pi$ changes a little and becomes $\pi'$, and $\delta_a = D_{\text{TV}}(\pi(s), \pi'(s))$, then we have

$$\left| \sum_{t=1}^{\infty} \gamma^t \left( \rho^\pi(s, \pi(s), t) - \rho^\pi(s, \pi'(s), t) \right) \right|$$

$$= \left| \mathbb{E}_{a_1 \sim \pi} \sum_{t=1}^{\infty} \gamma^t \rho^\pi(s, a_1, t) - \mathbb{E}_{a_2 \sim \pi'} \sum_{t=1}^{\infty} \gamma^t \rho^\pi(s, a_1, t) \right| \tag{7}$$

$$\leq \epsilon_\pi \delta_a$$

This inequality comes from Lemma 1.

We denote the difference between $d_2^\pi(s)$ and $d_2^{\pi'}(s)$ as $\Delta d_2(s)$, which can be bounded as follows:

$$\Delta d_2(s) = |d_2^\pi(s) - d_2^{\pi'}(s)|$$

$$= \left| \sum_{t=1}^{\infty} \gamma^t \left( \rho^\pi(s, \pi(s), t) - \rho^\pi(s, \pi'(s), t) \right) d_1^\pi(s) \right|$$

$$= d_1^\pi(s) \left| \sum_{t=1}^{\infty} \gamma^t \left( \rho^\pi(s, \pi(s), t) - \rho^\pi(s, \pi'(s), t) \right) \right|$$

$$\leq \epsilon_\pi \delta_a d_1^\pi(s)$$

Recursively, we have

$$\Delta d_i(s) \leq \epsilon_\pi^{i-1} \delta_a^{i-1} d_1^\pi(s)$$

Obviously, the change of $\pi$ at state $s$ won't change $d_1^\pi(s)$. According to Eq. (3),

$$\Delta d(s) \leq \sum_{i=1}^{\infty} \Delta d_i(s)$$

$$\leq \sum_{i=2}^{\infty} (\epsilon_\pi \delta_a)^{i-1} d_1^\pi(s)$$

$$= \frac{\epsilon_\pi \delta_a}{1 - \epsilon_\pi \delta_a} d_1^\pi(s)$$

According to $\frac{\partial d^\pi(s)}{\partial \pi(s)} = \lim_{\delta_a \to 0} \frac{\Delta d(s)}{\delta_a}$, we have

$$\left| \frac{\partial d^\pi(s)}{\partial \pi(s)} \right| \leq \epsilon_\pi d_1^\pi(s)$$

This concludes the proof. $\qquad \square$

**Lemma 3.** *Given two policy $\pi_1$ and $\pi_2$, where $\pi_1(a|s) = \frac{\exp(Q_1(s,a))}{\sum_{a'} \exp(Q_1(s,a'))}$. Then*

$$\mathbb{E}_{a \sim \pi_2} Q_1(s, a) - \mathbb{E}_{a \sim \pi_1} Q_1(s, a) \leq 1$$

*Proof.* Suppose there are two actions $a_1$, $a_2$ under state $s$, and let $Q_1(s, a_1) = u$, $Q_1(s, a_2) = v$. Without loss of generality, let $u \leq v$.

$$\mathbb{E}_{a \sim \pi_2} Q_1(s, a) - \mathbb{E}_{a \sim \pi_1} Q_1(s, a) \leq v - \frac{u e^u + v e^v}{e^u + e^v}$$

$$= v - \frac{u + v e^{v-u}}{1 + e^{v-u}}$$

$$= v - u - \frac{(v - u) e^{v-u}}{1 + e^{v-u}}$$

Let $f(z) = z - \frac{z e^z}{1 + e^z}$, the maximum point $z_0$ of $f(z)$ satisfies $f'(z_0) = 0$ where $f'$ is the derivative of $f$, i.e., $\frac{e^{z_0}(1 + z_0 + e^{z_0})}{(1 + e^{z_0})^2} - 1 = 0$. This implies $1 + e^{z_0} = z_0 e^{z_0}$ and $z_0 \in (1, 2)$. We have

$$\mathbb{E}_{a\sim\pi_2}Q_1(s,a) - \mathbb{E}_{a\sim\pi_1}Q_1(s,a) \le f(v-u) \le z_0 - 1 \le 1$$

If the number of action is more than 2 and $Q_1(s,a_1) \ge Q_1(s,a_2) \ge \cdots Q_1(s,a_n)$, let $b_1$ represents $a_1$ and $b_2$ represents all other actions. Then $Q_1(s,b_1) = Q_1(s,a_1)$ and $Q_1(s,b_2) = \sum_{j=2}^{n} \frac{Q_1(s,a_j)\exp(Q_1(s,a_j))}{\sum_{k=2}^{n}\exp(Q_1(s,a_k))}$. In this way, we can derive the upper bound of $\mathbb{E}_{a\sim\pi_2}Q_1(s,a) - \mathbb{E}_{a\sim\pi_1}Q_1(s,a)$ as above. $\qquad\square$

The following lemma is proposed by Kakade,

**Lemma 4** (Lemma 6.1 in [1])**.** *For any policy $\tilde\pi$ and $\pi$,*

$$\eta(\tilde\pi) - \eta(\pi) = \frac{1}{1-\gamma}\mathbb{E}_{d^{\tilde\pi}(s,a)}[A_\pi(s,a)] \tag{8}$$

**Lemma 5.** *In discrete MDPs, let $\epsilon_{\pi_k} = \sup_{s,a}\sum_{t=1}^{\infty}\gamma^t\rho^{\pi_k}(s,a,t)$, the optimal solution $p_k$ to a relaxation of optimization problem 2 satisfies the following relationship:*

$$p_k(s,a) = \frac{1}{Z^*}\left(D_k(s,a) + \epsilon_k(s,a)\right) \tag{9}$$

*where $D_k(s,a) = d^{\pi_k}(s,a)(2-\pi_k(a|s))\exp\left(-|Q_k - Q^*|\,(s,a)\right)|Q_k - \mathcal{B}^*Q_{k-1}|\,(s,a)$, $Z^*$ is the normalization constant and $\frac{\epsilon_k(s,a)}{D_k(s,a)} \le \epsilon_{\pi_k}$.*

*Proof.* Suppose $a^* \sim \pi^*(s)$. Let $\pi = \pi_k, \tilde\pi = \pi^*$ in Lemma 4, we have

$$
\begin{aligned}
&\eta(\pi^*) - \eta(\pi_k) \\
&= -\frac{1}{1-\gamma}\mathbb{E}_{d^{\pi_k}(s,a)}A_{\pi^*}(s,a) \\
&= \frac{1}{1-\gamma}\mathbb{E}_{d^{\pi_k}(s,a)}(V^*(s) - Q^*(s,a)) \\
&= \frac{1}{1-\gamma}\mathbb{E}_{d^{\pi_k}(s,a)}\Big(V^*(s) - Q_k(s,a^*) + Q_k(s,a^*) - Q_k(s,a) + Q_k(s,a) - Q^*(s,a)\Big) \\
&\overset{(a)}{\le} \frac{1}{1-\gamma}\Big(\mathbb{E}_{d^{\pi_k}(s)}(Q^*(s,a^*) - Q_k(s,a^*)) + \mathbb{E}_{d^{\pi_k}(s,a)}(Q_k(s,a) - Q^*(s,a)) + 1\Big) \\
&\le \frac{1}{1-\gamma}\Big(\mathbb{E}_{d^{\pi_k}(s)}|Q^*(s,a^*) - Q_k(s,a^*)| + \mathbb{E}_{d^{\pi_k}(s,a)}|Q_k(s,a) - Q^*(s,a)| + 1\Big) \\
&= \frac{2}{1-\gamma}\Big(\mathbb{E}_{d^{\pi_k,\pi^*}}|Q_k(s,a) - Q^*(s,a)| + 1\Big),
\end{aligned}
\tag{10}
$$

where $d^{\pi_k,\pi^*}(s,a) = d^{\pi_k}(s)\frac{\pi_k(a|s)+\pi^*(a|s)}{2}$ and (a) uses Lemma 3.

Since both sides of the above equation have the same minimum (here the minima are given by $Q_k = Q^*$), we can replace the objective in Problem 2 with the upper bound in Eq. (10) and solve the relaxed optimization problem.

$$\min_{p_k} \quad \mathbb{E}_{d^{\pi_k}(s,a)}[|Q_k - Q^*|] \tag{11}$$

$$\text{s.t.} \quad Q_k = \arg\min_{Q\in\mathcal{Q}}\mathbb{E}_{p_k}[(Q - \mathcal{B}^\pi Q_{k-1})^2(s,a)], \tag{12}$$

$$\sum_{s,a}p_k(s,a) = 1, \quad p_k(s,a) \ge 0. \tag{13}$$

Here we use $d^{\pi_k}(s,a)$ to replace $d^{\pi_k,\pi^*}$ because we can not access $\pi^*$, and the best surrogate available is $\pi_k$.

**Step 1: Jensen's Inequality.** The optimization objective can be further relaxed with Jensen's Inequality, based on the fact that $f(x) = \exp(-x)$ is a convex function.

$$\mathbb{E}_{d^{\pi_k}(s,a)}[|Q_k - Q^*|] = -\log\exp(-\mathbb{E}_{d^{\pi_k}(s,a)}[|Q_k - Q^*|]) \le -\log\mathbb{E}_{d^{\pi_k}(s,a)}[\exp(-|Q_k - Q^*|)] \tag{14}$$

Similarly, both sides of Eq. (14) have the same minimum. We obtain the following new optimization problem by replacing the objective with the upper bound in this equation:

$$\min_{p_k} - \log \mathbb{E}_{d^{\pi_k}(s,a)}[\exp(-|Q_k - Q^*|)]$$

$$\text{s.t.} \quad Q_k = \arg\min_{Q \in \mathcal{Q}} \mathbb{E}_{p_k}[(Q - \mathcal{B}^* Q_{k-1})^2], \tag{15}$$

$$\sum_{s,a} p_k(s,a) = 1, \quad p_k(s,a) \geq 0.$$

**Step 2: Computing the Lagrangian.** In order to optimize problem 15, we follow the standard procedures of Lagrangian multiplier method. The Lagrangian is:

$$\mathcal{L}(p_k; \lambda, \mu) = -\log \mathbb{E}_{d^{\pi_k}(s,a)}[\exp(-|Q_k - Q^*|)] + \lambda(\sum_{s,a} p_k(s,a) - 1) - \mu^T p_k. \tag{16}$$

where $\lambda$ and $\mu$ are the Lagrange multipliers.

**Step 3: IFT gradient used in the Lagrangian.** $\frac{\partial Q_k}{\partial p_k}$ can be computed according to implicit function theorem (IFT). The IFT gradient is given by:

$$\left. \frac{\partial Q_k}{\partial p_k} \right|_{Q_k, p_k} = - \left[ \text{Diag}\,(p_k) \right]^{-1} \left[ \text{Diag}\,(Q_k - \mathcal{B}^* Q_{k-1}) \right] \tag{17}$$

The derivation is similar to that in [2].

**Step 4: Approximation of the gradient used in the Lagrangian.** We derive an expression for $\frac{\partial d^{\pi_k}(s,a)}{\partial p_k}$, which will be used when computing the gradient of the Lagrangian. We use $\pi_k$ to denote the policy induced by $Q_k$.

$$\frac{\partial d^{\pi_k}(s,a)}{\partial p_k} = \frac{\partial d^{\pi_k}(s,a)}{\partial \pi_k} \frac{\partial \pi_k}{\partial Q_k} \frac{\partial Q_k}{\partial p_k}$$

$$= (d^{\pi_k}(s) + \epsilon_2(s)) \frac{\partial \pi_k}{\partial Q_k} \frac{\partial Q_k}{\partial p_k}$$

$$\overset{(b)}{=} (d^{\pi_k}(s) + \epsilon_2(s) \pi_k(a|s) \frac{\sum_{a' \neq a} \exp(Q_k(s,a'))}{\sum_{a'} \exp(Q_k(s,a'))} \frac{\partial Q_k}{\partial p_k}$$

$$\overset{(c)}{=} d^{\pi_k}(s,a)(1 - \pi_k(a|s)) \frac{\partial Q_k}{\partial p_k} + \epsilon_2(s) \pi_k(a|s)(1 - \pi_k(a|s)) \frac{\partial Q_k}{\partial p_k}$$

where $\epsilon_2(s) = \frac{\partial d^{\pi_k}(s)}{\partial \pi_k(s)}$. (b) and (c) are based on the fact that $\pi_k(a|s) = \frac{\exp(Q_k(s,a))}{\sum_{a'} \exp(Q_k(s,a'))}$.

**Step 5: Computing optimal $p_k$.** By KKT conditions, we have

$$\frac{\partial \mathcal{L}(p_k; \lambda, \mu)}{\partial p_k} = 0$$

$$\frac{\partial \mathcal{L}(p_k; \lambda, \mu)}{\partial p_k}$$

$$= \frac{\exp(-|Q_k - Q^*|(s,a))}{Z} (d^{\pi_k}(s,a) \text{sgn}(Q_k - Q^*) \cdot \frac{\partial Q_k}{\partial p_k} + \cdot \frac{\partial d^{\pi_k}(s,a)}{\partial p_k}) + \lambda - \mu_{s,a}$$

where $Z = \mathbb{E}_{s',a' \sim d^{\pi_k}(s,a)} \exp(-|Q_k - Q^*|(s',a'))$. Substituting the expression of $\frac{\partial Q_k}{\partial p_k}$ and $\frac{\partial d^{\pi_k}(s,a)}{\partial p_k}$ with the results obtained in Step. 3 and Step. 4 respectively, and let $Z_{s,a} = Z(\lambda^* - \mu_{s,a}^*)$, we obtain

$$p_k(s,a) = \Big( d^{\pi_k}(s,a)(\text{sgn}(Q_k - Q^*) + 1 - \pi_k(a|s)) \exp(-|Q_k - Q^*|(s,a)) |Q_k - \mathcal{B}^* Q_{k-1}|(s,a)$$

$$+ \epsilon_2(s) \pi_k(a|s)(1 - \pi_k(a|s)) \exp(-|Q_k - Q^*|(s,a)) |Q_k - \mathcal{B}^* Q_{k-1}|(s,a) \Big) \frac{1}{Z_{s,a}}$$

$$\tag{18}$$

Notably, $Q_k \approx Q^{\pi_k} \le Q^*$. Thus, $\text{sgn}(Q_k - Q^*)$ always is 1 approximately, so we can simplify this relationship as

$$p_k(s,a) = \frac{1}{Z_{s,a}} \Big( d^{\pi_k}(s,a)(2 - \pi_k(a|s)) \exp\left(-|Q_k - Q^*|(s,a)\right) |Q_k - \mathcal{B}^* Q_{k-1}|(s,a)$$
$$+ \epsilon_2(s)\pi_k(a|s)(1 - \pi_k(a|s)) \exp\left(-|Q_k - Q^*|(s,a)\right) |Q_k - \mathcal{B}^* Q_{k-1}|(s,a) \Big) \tag{19}$$

The first term is always larger or equal to zero. The second term does not influence the sign of the equation because the absolute value of $\epsilon_2(s)$ is smaller than $d^{\pi_k}(s)$ according to Lemma 2. Note that Eq. (19) is always larger or equal to zero. If it is larger than zero then $\mu^* = 0$ by the KKT condition. If it is equal to zero, we can let $\mu^* = 0$ because the value of $\mu^*$ does not influence $w_k(s,a)$. Without loss of generality, we can let $\mu^* = 0$. Then $Z_{s,a} = Z^* = Z\lambda^*$ is a constant with respect to different $s$ and $a$. In this way, we can simplify Eq. (19) as follows:

$$p_k(s,a) = \frac{1}{Z^*}\left(D_k(s,a) + \epsilon_k(s,a)\right)$$

where $D_k(s,a) = d^{\pi_k}(s,a)(2 - \pi_k(a|s)) \exp\left(-|Q_k - Q^*|(s,a)\right) |Q_k - \mathcal{B}^* Q_{k-1}|(s,a)$ and $\epsilon_k(s,a) = \epsilon_2(s)\pi_k(a|s)(1 - \pi_k(a|s)) \exp\left(-|Q_k - Q^*|(s,a)\right) |Q_k - \mathcal{B}^* Q_{k-1}|(s,a)$.

Based on the expression of $D_k(s,a)$ and $\epsilon_k(s,a)$, we have

$$\frac{\epsilon_k(s,a)}{D_k(s,a)} = \frac{\epsilon_2(s)(1 - \pi_k(a|s))}{d^{\pi_k}(s)(2 - \pi_k(a|s))} \le \epsilon_{\pi_k}$$

The inequality is from 2. This concludes the proof. □

**Theorem 1** (formal). *Let* $\epsilon_{\pi_k} = \sup_{s,a} \sum_{t=1}^{\infty} \gamma^t \rho^{\pi_k}(s,a,t)$. *Under Assumption 1, 2 and 3, if* $\frac{d^{\pi_k}(s,a)}{\mu(s,a)}$ *exists, we have in MDPs with discrete action spaces, the solution* $w_k$ *to the relaxed optimization problem 1 is*

$$w_k(s,a) = \frac{1}{Z_1^*}\left(E_k(s,a) + \epsilon_{k,1}(s,a)\right). \tag{20}$$

*In MDPs with continuous action spaces, the solution is*

$$w_k(s,a) = \frac{1}{Z_2^*}\left(F_k(s,a) + \epsilon_{k,2}(s,a)\right). \tag{21}$$

*where*

$$E_k(s,a) = \frac{d^{\pi_k}(s,a)}{\mu(s,a)}(2 - \pi_k(a|s)) \exp\left(-|Q_k - Q^*|(s,a)\right) |Q_k - \mathcal{B}^* Q_{k-1}|(s,a)$$

$$F_k(s,a) = 2\frac{d^{\pi_k}(s,a)}{\mu(s,a)} \exp\left(-|Q_k - Q^*|(s,a)\right) |Q_k - \mathcal{B}^* Q_{k-1}|(s,a),$$

$Z_1^*$, $Z_2^*$ *is the normalization constants and* $\max\left\{\frac{\epsilon_{k,1}(s,a)}{E_k(s,a)}, \frac{\epsilon_{k,2}(s,a)}{F_k(s,a)}\right\} \le \epsilon_{\pi_k}$.

*Proof.* By Lemma 5, for MDPs with discrete action space and state space, we have

$$p_k(s,a) = \frac{1}{Z^*}\left(D_k(s,a) + \epsilon_k(s,a)\right)$$

Based on the deviation of Problem 2, the solution in this situation is

$$w_k(s,a) = \frac{1}{Z^*}\left(\frac{D_k(s,a)}{\mu(s,a)} + \frac{\epsilon_k(s,a)}{\mu(s,a)}\right) \tag{22}$$

The existence of $\frac{d^{\pi_k}(s,a)}{\mu(s,a)}$ guarantees the existence of $\frac{D_k(s,a)}{\mu(s,a)}$ and $\frac{\epsilon_k(s,a)}{\mu(s,a)}$. Let $E_k(s,a) = \frac{D_k(s,a)}{\mu(s,a)}$ and $\epsilon_{k,1}(s,a) = \frac{\epsilon_k(s,a)}{\mu(s,a)}$, we get Eq. (20).

We derive the result for continuous action space and state space as follows, the result for continuous state space and discrete action space, and discrete state space and continuous action space can be derived similarly.

Remember that $\mathcal{B}^*Q_{k-1}(s,a) = r(s,a) + \gamma\max_{a'}\mathbb{E}_{s'}Q_{k-1}(s',a')$ and $Q_k(s,a) = \arg\min_Q(Q(s,a) - \mathcal{B}^*Q_{k-1}(s,a))^2$, if we use $R(s,a) = Q_k(s,a) - \gamma\max_{a'}\mathbb{E}_{s'}Q_{k-1}(s',a')$ to replace $r(s,a)$, then $Q_k$ is still the desired Q function after the Bellman update. Since the continuity of $Q_k$, $Q_{k-1}$ and $T$ guarantee $R(s,a)$ is continuous, without loss of generality, we assume $r(s,a)$ is continuous.

We utilize the techniques in reinforcement learning with aggregated states [3]. Concretely, we can partition the set of all state-action pairs, with each cell representing an aggregated state. Such a partition can be defined by a function $\phi : \mathcal{S} \cup \mathcal{A} \mapsto \hat{\mathcal{S}} \cup \hat{\mathcal{A}}$, where $\hat{\mathcal{S}}$ is the space of aggregated states and $\hat{\mathcal{A}}$ is the space of aggregated actions. With such a partition, we can discretize the continuous spaces. For example, for the continuous space $\{x \in \mathbb{R} : 0 \le x \le 10\}$, define $\phi(x) = \sum_{i=1}^{9}\mathbb{I}(x \le x_i)$, and then the space of aggregated states becomes $\{0, 1, 2, \ldots, 9\}$, which is a discrete space.

With function $\phi$, we define the transition function and reward function in this new MDP. For all $\hat{s}, \hat{s}' \in \hat{\mathcal{S}}, \hat{a} \in \hat{\mathcal{A}}$

$$\hat{T}(\hat{s}'|\hat{s},\hat{a}) = \frac{\sum_{s,a\in\phi^{-1}(\hat{s},\hat{a})}\mu(s,a)\sum_{s'\in\phi^{-1}(\hat{s}')}T(s'|s,a)}{\sum_{s,a\in\phi^{-1}(\hat{s},\hat{a})}\mu(s,a)}$$

$$\hat{r}(\hat{s},\hat{a}) = \frac{\sum_{s,a\in\phi^{-1}(\hat{s},\hat{a})}\mu(s,a)r(s,a)}{\sum_{s,a\in\phi^{-1}(\hat{s},\hat{a})}\mu(s,a)} \tag{23}$$

where $(\phi(s), \phi(a))$ is simplified as $\phi(s,a)$ and $\phi^{-1}(\hat{s},\hat{a})$ is the preimage of $(\hat{s},\hat{a})$.

In this way, Eq. (22) holds for aggregated state space:

$$\hat{w}_k(\phi(s,a)) = \frac{1}{\hat{Z}^*}\left(\frac{\hat{D}_k(\phi(s,a))}{\hat{\mu}(\phi(s,a))} + \frac{\hat{\epsilon}_k(\phi(s,a))}{\hat{\mu}(\phi(s,a))}\right) \tag{24}$$

Suppose $\hat{\mathcal{S}}$ and $\hat{\mathcal{A}}$ is equipped with metric $m'$, we construct a sequence of functions $\phi_h$, which satisfies

(i) If $m(u_1 - u_2) \le m(u_1 - u_3)$, then $m'(\phi_h(u_1) - \phi_h(u_2)) \le m'(\phi_h(u_1) - \phi_h(u_3))$ for all $u_1, u_2, u_3 \in \mathcal{S}$ or $u_1, u_2, u_3 \in \mathcal{A}$.

(ii) $\lim_{h\to\infty}\text{diam}(\phi_h^{-1}(c)) = 0$ for all $c \in \mathcal{S}' \cup \mathcal{A}'$.

Based on the two conditions on $\phi_h$ and the continuous of reward function and transition function, for all $s, s' \in \mathcal{S}$ and $a \in \mathcal{A}$,

$$\lim_{h\to\infty}|\hat{r}(\phi_h(s,a)) - r(s,a)| = 0$$

$$\lim_{h\to\infty}\left|\hat{T}(\phi_h(s')|\phi_h(s,a)) - T(s'|s,a)\right| = 0 \tag{25}$$

This means the constructed MDP approaches the original MDP as $h$ tends to infinity.

With the Lemma 3 in [4],

$$\lim_{h\to\infty}\mathcal{B}^*\hat{Q}_{k-1}(\phi_h(s,a)) = \mathcal{B}^*Q_{k-1}(s,a)$$

$$\lim_{h\to\infty}\mathcal{B}^*\hat{Q}^*(\phi_h(s,a)) = \mathcal{B}^*Q^*(s,a)$$

Note that $Q_k(s,a) = \arg\min_Q(Q - \mathcal{B}^*Q_{k-1}(s,a))^2$, $\hat{Q}_k(\phi_h(s,a)) = \arg\min_Q(Q - \mathcal{B}^*Q_{k-1}(\phi_h(s,a)))^2$, $Q^*(s,a) = \arg\min_Q(Q - \mathcal{B}^*Q^*(s,a))^2$ and $\hat{Q}^*(\phi_h(s,a)) = \arg\min_Q(Q -$

$\mathcal{B}^*Q^*(\phi_h(s,a)))^2$,

$$\lim_{h\to\infty} \hat{Q}_k(\phi_h(s,a)) = Q_k(s,a)$$

$$\lim_{h\to\infty} \hat{Q}^*(\phi_h(s,a)) = Q^*(s,a)$$

Because $\pi(a|s) = \frac{\exp(Q(s,a))}{\sum_{a'}\exp(Q(s,a'))}$, $\pi$ is continuous with respect to $Q$, then we have

$$\lim_{h\to\infty} \hat{\pi}_k(\phi_h(a)|\phi_h(s)) = \pi_k(a|s)$$

The continuity of $\pi$ and transition function $T$ guarantees

$$\lim_{h\to\infty} \hat{d}^{\hat{\pi}_k}(\phi_h(s,a)) = d^{\pi_k}(s,a)$$

Therefore,

$$\lim_{h\to\infty} |\hat{Q}_k - \hat{Q}^*|(\phi((s,a))) = |Q_k - Q^*|(s,a)$$

$$\lim_{h\to\infty} |\hat{Q}_k - \hat{\mathcal{B}}^*\hat{Q}^*|(\phi((s,a))) = |Q_k - \mathcal{B}^*Q_{k-1}|(s,a) \tag{26}$$

$$\lim_{h\to\infty} \frac{d^{\hat{\pi}_k}(\phi_h(s,a))}{\hat{\mu}(\phi_h(s,a))} = \frac{d^{\pi_k}(s,a)}{\mu(s,a)}$$

Notably, $\epsilon_2(s)\pi_k(a|s) \le d^{\pi_k}(s,a)$, the existence of $\frac{d^{\pi_k}(s,a)}{\mu(s,a)}$ implies the existence of $\frac{\epsilon_2(s)\pi_k(a|s)}{\mu(s,a)}$.

$$\lim_{h\to\infty} \frac{\hat{\epsilon}_k(\phi(s,a))}{\hat{\mu}(\phi(s,a))} = \epsilon_{k,1}(s,a) \tag{27}$$

where $\epsilon_{k,1} = \frac{\epsilon_k(s)\pi_k(a|s)}{\mu(s,a)}(1 - \pi_k(a|s))\exp\left(-|Q_k - Q^*|(s,a)\right)|Q_k - \mathcal{B}^*Q_{k-1}|(s,a)$.

Using the Eq. (A), (26) and (27), we have

$$w_k(s,a) = \frac{1}{Z_1^*}\left(E_k(s,a) + \epsilon_{k,1}(s,a)\right).$$

If the action space is continuous, $\pi_k(a|s) = 0$, then we have

$$w_k(s,a) = \frac{1}{Z_2^*}\left(F_k(s,a) + \epsilon_{k,2}(s,a)\right)$$

The upper bound of $\frac{\epsilon_{k,1}(s,a)}{E_k(s,a)}$ and $\frac{\epsilon_{k,2}(s,a)}{F_k(s,a)}$ can be derived directly from Lemma 5. This concludes our proof. $\qquad\square$

## B   Detailed Proof of Theorem 2

Let $(\mathcal{B}Q)_k(s,a)$ denote $|Q_k(s,a) - \mathcal{B}^*Q_k(s,a)|$. We first introduce an assumption.

**Assumption 4.** *At iteration $k$, $(\mathcal{B}Q)_k(s,a)$ is independent of $(\mathcal{B}Q)_k(s',a')$ if $(s,a) \ne (s',a')$ for all $k > 0$.*

This assumption is not strong. If we use a table to represent Q function, it holds apparently. Notably, though we need this assumption in our proof, we can also apply our method on the situation where this assumption doesn't hold. With this assumption, we have the following theorem.

**Lemma 6.** *Consider a MDP, trajectories $\tau_i = \{s_t^i, a_t^i\}_{t=0}^{T_i}$, $i = 0, 1, \ldots$ is generated by a policy $\pi$ under this MDP, then we have*

$$|Q_k(s,a) - Q^*(s,a)| \le |Q_k(s_t,a_t) - \mathcal{B}^*Q_{k-1}(s_t,a_t)|$$

$$+ \mathbb{E}_\tau\left(\sum_{t'=1}^{h_\tau^{\pi_k}(s,a)} \gamma^{t'}\left((\mathcal{B}Q)_{k-1}(s_{t'},a_{t'}) + c\right) + \gamma^{h_\tau^{\pi_k}(s,a)+1}c\right) \tag{28}$$

where $(\mathcal{B}Q)_k(s_{h_\tau^{\pi_k}(s,a)}, a_{h_\tau^{\pi_k}(s,a)}) = |Q_k(s_{h_\tau^{\pi_k}(s,a)}, a_{h_\tau^{\pi_k}(s,a)}) - r(s_{h_\tau^{\pi_k}(s,a)}, a_{h_\tau^{\pi_k}(s,a)})|$, $c = \max_{s,a}\left(Q^*(s,a^*) - Q^*(s,a)\right)$, and $(s_{t'}, a_{t'})$ is the $t'$-th state-action pair behind $(s,a)$.

*Proof.*

$$
\begin{aligned}
&|Q_k(s_t, a_t) - Q^*(s_t, a_t)| \\
&= |Q_k(s_t, a_t) - \mathcal{B}^*Q_{k-1}(s_t, a_t) + \mathcal{B}^*Q_{k-1}(s_t, a_t) - \mathcal{B}^*Q^*(s_t, a_t)]| \\
&\overset{(a)}{\le} |Q_k(s_t, a_t) - \mathcal{B}^*Q_{k-1}(s_t, a_t)| \\
&\quad + \gamma|\mathbb{E}_{p(\tau)}[Q_{k-1}(s_{t+1}, a_{t+1}) - Q^*(s_{t+1}, a_{t+1}) + Q^*(s_{t+1}, a_{t+1}) - Q^*(s_{t+1}, a^*)]| \\
&\overset{(b)}{\le} |Q_k(s_t, a_t) - \mathcal{B}^*Q_{k-1}(s_t, a_t)| + \gamma c + \gamma\mathbb{E}_\tau[|Q_{k-1}(s_{t+1}, a_{t+1}) - Q^*(s_{t+1}, a_{t+1})|]
\end{aligned}
$$

where the expectation is taken over $s' \sim P(s'|s,a)$, $a' \sim \pi(a'|s')$. (a) uses triangle inequality, (b) is because $f(x) = |x|$ is convex function and using Jensen's Inequality.

Similarly, we have

$$
\begin{aligned}
&|Q_{k-1}(s_{t+1}, a_{t+1}) - Q^*(s_{t+1}, a_{t+1})| \\
&= |Q_{k-1}(s_{t+1}, a_{t+1}) - \mathcal{B}^*Q_{k-1}(s_{t+1}, a_{t+1}) + \mathcal{B}^*Q_{k-1}(s_{t+1}, a_{t+1}) - \mathcal{B}^*Q^*(s_{t+1}, a_{t+1})]| \\
&\le (\mathcal{B}Q)_{k-1}(s_{t+1}, a_{t+1}) + \gamma c + \gamma\mathbb{E}_\tau[|Q_{k-1}(s_{t+2}, a_{t+2}) - Q^*(s_{t+2}, a_{t+2})|]
\end{aligned}
$$

Recursively,

$$
\begin{aligned}
&|Q_k(s,a) - Q^*(s,a)| \\
&\le |Q_k(s_t, a_t) - \mathcal{B}^*Q_{k-1}(s_t, a_t)| + \sum_{t'=1}^{h_\tau^{\pi_k}(s,a)} \gamma^{t'}\left((\mathcal{B}Q)_{k-1}(s_{t'}, a_{t'}) + c\right) + \gamma^{h_\tau^{\pi_k}(s,a)+1}c
\end{aligned}
\tag{29}
$$

where $(\mathcal{B}Q)_{k-1}(s_{h_\tau^{\pi_k}(s,a)}, a_{h_\tau^{\pi_k}(s,a)}) = |Q_{k-1}(s_{h_\tau^{\pi_k}(s,a)}, a_{h_\tau^{\pi_k}(s,a)}) - r(s_{h_\tau^{\pi_k}(s,a)}, a_{h_\tau^{\pi_k}(s,a)})|$. $\square$

This theorem shows that the cumulative Bellman error with a constant $c$ is an upper bound of $|Q_k - Q^*|$, so we can use Bellman error with the constant to estimate this quantity.

Suppose the Q function is equipped with a learning rate $\alpha$, i.e., $Q_k = \alpha(\mathcal{B}^*Q_{k-1} - Q_{k-1}) + (1 - \alpha)Q_{k-1}$, we have the following lemma,

**Lemma 7.**

$$
\begin{aligned}
\|\mathcal{B}^*Q_k - Q_k\|_\infty &\le (\alpha\gamma + 1 - \alpha)^k \|\mathcal{B}^*Q_0 - Q_0\|_\infty \\
\|\mathcal{B}^*Q_{k-1} - Q_k\|_\infty &\le (1 - \alpha)(\alpha\gamma + 1 - \alpha)^{k-1} \|\mathcal{B}^*Q_0 - Q_0\|_\infty
\end{aligned}
\tag{30}
$$

*Proof.*

$$
\begin{aligned}
Q_k &= Q_{k-1} + \alpha(\mathcal{B}^*Q_{k-1} - Q_{k-1}) \\
&\implies \mathcal{B}^*Q_{k-1} - Q_k = \frac{1-\alpha}{\alpha}(Q_k - Q_{k-1})
\end{aligned}
$$

$$
\begin{aligned}
\|\mathcal{B}^*Q_k - Q_k\|_\infty &\le \|\mathcal{B}^*Q_k - \mathcal{B}^*Q_{k-1}\|_\infty + \|\mathcal{B}^*Q_{k-1} - Q_k\|_\infty \\
&\le \gamma\|Q_k - Q_{k-1}\|_\infty + \|\mathcal{B}^*Q_{k-1} - Q_k\|_\infty \\
&\le (\gamma + \frac{1-\alpha}{\alpha})\|Q_k - Q_{k-1}\|_\infty \\
&\le (\alpha\gamma + 1 - \alpha)\|\mathcal{B}^*Q_{k-1} - Q_{k-1}\|_\infty
\end{aligned}
\tag{31}
$$

$$
\begin{aligned}
\|Q_k - \mathcal{B}^*Q_{k-1}\|_\infty &\le (1 - \alpha)\|Q_{k-1} - \mathcal{B}^*Q_{k-1}\|_\infty \\
&\le (1 - \alpha)\left(\|Q_{k-1} - \mathcal{B}^*Q_{k-2}\|_\infty + \|\mathcal{B}^*Q_{k-2} - \mathcal{B}^*Q_{k-1}\|_\infty\right) \\
&\le (1 - \alpha)\left(\gamma\|Q_{k-2} - Q_{k-1}\|_\infty + \|Q_{k-1} - \mathcal{B}^*Q_{k-2}\|_\infty\right) \\
&\overset{(a)}{\le} (1 - \alpha)(\gamma + \frac{1-\alpha}{\alpha})\|Q_{k-1} - Q_{k-2}\|_\infty \\
&\le (1 - \alpha)(\alpha\gamma + 1 - \alpha)\|\mathcal{B}^*Q_{k-2} - Q_{k-2}\|_\infty
\end{aligned}
\tag{32}
$$

Then we can finish the proof by recursively applying Eq. (31) and (32). □

**Lemma 8** (Azuma). *Let $X_0, X_1, \ldots$ be a martingale such that, for all $k \geq 1$, $|X_k - X_{k-1}| \leq c_k$, Then*

$$\Pr[|X_n - X_0| \geq t] \leq 2\exp(-\frac{t^2}{2\sum_{k=1}^n c_k^2}). \tag{33}$$

In the follows, we denote $\sum_{t=1}^{h_\tau^{\pi_k}(s,a)} \gamma^t (\mathcal{B}Q)_k(s_t, a_t)$ as $\mathcal{B}(s, a, k)$.

**Lemma 9.** *Let $\phi_k = (\alpha\gamma + 1 - \alpha)^k \|\mathcal{B}^* Q_0 - Q_0\|_\infty$, $f(t) = \frac{\gamma - \gamma^{t+1}}{1-\gamma}$ and $\epsilon_{\pi_k} = \sup_{s,a} \sum_{t=1}^\infty \gamma^t \rho^{\pi_k}(s, a, t)$. Under Assumption 4, with probability at least $1 - \delta$,*

$$|\mathcal{B}(s, a, k) - f(h_\tau^{\pi_k}(s,a))\mathbb{E}[(\mathcal{B}Q)_k(s_t, a_t)]| \leq \sqrt{2f(h_\tau^{\pi_k}(s,a))^2(1 + \epsilon_{\pi_k})^2 \phi_k^2 \log\frac{2}{\delta}}. \tag{34}$$

*Proof.* Let $\mathcal{F}_h = \sigma_t(s_0, a_0, r_0, \ldots, s_{h-1}, a_{h-1}, r_{h-1})$ be the $\sigma$-field summarising the information available just before $s_t$ is observed.

Define $Y_h = \mathbb{E}[\mathcal{B}(s, a, k)|\mathcal{F}_h]$, then $Y_h$ is a martingale because

$$\mathbb{E}[Y_h|\mathcal{F}_{h-1}] = \mathbb{E}[\mathbb{E}[\mathcal{B}(s, a, k)|\mathcal{F}_h]|\mathcal{F}_{h-1}] = \mathbb{E}[\mathcal{B}(s, a, k)|\mathcal{F}_{h-1}] = Y_{h-1}$$

$$\begin{aligned}
|Y_h - Y_{h-1}| &\leq \gamma^h (1 + \epsilon_{\pi_k}) \|\mathcal{B}^* Q_k - Q_k\|_\infty \\
&\leq \gamma^h (1 + \epsilon_{\pi_k})(\alpha\gamma + 1 - \alpha)^k \|\mathcal{B}^* Q_0 - Q_0\|_\infty = \gamma^h (1 + \epsilon_{\pi_k})\phi_k
\end{aligned}$$

By Azuma's lemma,

$$\Pr\left(|\mathcal{B}(s, a, k) - \mathbb{E}[\mathcal{B}(s, a, k)]| \geq \sqrt{2\Big(\frac{\gamma - \gamma^{h_\tau^{\pi_k}+1}}{1-\gamma}\Big)^2 (1 + \epsilon_{\pi_k})^2 \phi_k^2 \log\frac{2}{\delta}}\right) \leq \delta$$

□

Since $(\alpha\gamma + 1 - \alpha)$ is less than 1, $\phi_k$ decreases exponentially as $k$ increases. This theorem shows that we can use the average Bellman error as a surrogate of Bellman error at specific state-action pair without losing too much accuracy. In this way, $|Q_k - Q^*|(s, a)$ is merely related to the distance to end of the state-action pair.

**Theorem 2** (formal). *Under Assumption 4, with probability at least $1 - \delta$, we have*

$$\begin{aligned}
&|Q_k(s, a) - Q^*(s, a)| \\
&\leq \mathbb{E}_\tau\left(f(h_\tau^{\pi_k}(s,a))\big(\mathbb{E}[(\mathcal{B}Q)_k(s_{t'}, a_{t'})] + c\big) + \gamma^{h_\tau^{\pi_k}(s,a)+1}c\right) + g(k, \delta)
\end{aligned} \tag{35}$$

*where $g(k, \delta) = (1 - \alpha)\phi_{k-1} + \sqrt{2f(h_\tau^{\pi_k}(s,a))^2(1 + \epsilon_{\pi_k})^2 \phi_k^2 \log\frac{2}{\delta}}$.*

*Proof.* According to Lemma 6, we have

$$\begin{aligned}
|Q_k(s, a) - Q^*(s, a)| \leq &|Q_k(s_t, a_t) - \mathcal{B}^* Q_{k-1}(s_t, a_t)| \\
&+ \mathbb{E}_\tau\left(\sum_{t'=1}^{h_\tau^{\pi_k}(s,a)} \gamma^{t'}\Big((\mathcal{B}Q)_{k-1}(s_{t'}, a_{t'}) + c\Big) + \gamma^{h_\tau^{\pi_k}(s,a)+1}c\right)
\end{aligned} \tag{36}$$

Using Lemma 7, we can upper bound $|Q_k(s_t, a_t) - \mathcal{B}^* Q_{k-1}(s_t, a_t)|$ as $(1 - \alpha)\phi_{k-1}$. With Lemma 9, $\sum_{t=1}^{h_\tau^{pi_k}(s,a)} \gamma^t (\mathcal{B}Q)_k(s_t, a_t)$ can be bounded by right hand side of Eq. (34) with probability $1 - \delta$.

Substitute the bounds into Eq. (36), we have

$$
\begin{aligned}
|Q_k(s,a) - Q^*(s,a)| \leq (1-\alpha)\phi_{k-1} &+ \sqrt{2f(h_\tau^{\pi_k}(s,a))^2(1+\epsilon_{\pi_k})^2\phi_k^2\log\frac{2}{\delta}} \\
&+ \mathbb{E}_\tau\left(f(h_\tau^{\pi_k}(s,a))\big(\mathbb{E}[(\mathcal{B}Q)_k(s_{t'},a_{t'})] + c\big) + \gamma^{h_\tau^{\pi_k}(s,a)+1}c\right) \\
\leq g(k,\delta) & \\
&+ \mathbb{E}_\tau\left(f(h_\tau^{\pi_k}(s,a))\big(\mathbb{E}[(\mathcal{B}Q)_k(s_{t'},a_{t'})] + c\big) + \gamma^{h_\tau^{\pi_k}(s,a)+1}c\right)
\end{aligned}
$$

$\square$

## C  Algorithms

---

**Algorithm 1** ReMERN

---

1: Initialize Q-values $Q_\theta(s,a)$, a replay buffer $\mu$, an error model $\Delta_\phi(s,a)$, and a weight model $\kappa_\psi$.

2: **for** step $k$ in $\{1,\ldots,N\}$ **do**
3:     Collect $M$ samples using $\pi_k$, add them to replay buffer $\mu$, sample $\{(s_i,a_i)\}_{i=1}^N \sim \mu$.
4:     Evaluate $Q_\theta(s,a)$, $\Delta_\phi(s,a)$ and $\kappa_\psi(s,a)$ on samples $(s_i,a_i)$.
5:     Compute target values for $Q$ and $\Delta$ on samples:
    $y_i = r_i + \gamma\max_{a'} Q_{k-1}(s_i',a')$.
    $\hat{a}_i = \arg\max_a Q_{k-1}(s_i',a)$.
    $\hat{\Delta} = |Q_\theta(s,a) - y_i| + \gamma\Delta_{k-1}(s_i',\hat{a}_i)$.
6:     Optimize $\kappa_\psi$ using

$$L_\kappa(\psi) := \mathbb{E}_{\mathcal{D}_s}\left[f^*\left(f'\left(\kappa_\psi(s,a)\right)\right)\right] - \mathbb{E}_{\mathcal{D}_f}\left[f'\left(\kappa_\psi(s,a)\right)\right].$$

7:     Compute $w_k$ using

$$w_k(s,a) \propto \frac{d^{\pi_k}(s,a)}{\mu(s,a)}\exp\left(-\gamma\left[P^{\pi^{w_{k-1}}}\Delta_{k-1}\right](s,a)\right).$$

8:     Minimize Bellman error for $Q_\theta$ weighted by $w_k$.
    $\theta_{k+1} \leftarrow \arg\min_\theta \frac{1}{N}\sum_i^N w_k(s_i,a_i)\left(Q_\theta(s_i,a_i) - y_i\right)^2$.
9:     Minimize ADP error for training $\phi$.
    $\phi_{k+1} \leftarrow \arg\min_\phi \frac{1}{N}\sum_{i=1}^N\left(\Delta_\phi(s_i,a_i) - \hat{\Delta}_i\right)^2$.
10: **end for**

---

**Algorithm 2** ReMERT

1: Initialize Q-values $Q_\theta(s, a)$, a replay buffer $\mu$, and a weight model $\kappa_\psi$.
2: **for** step $k$ in $\{1, \ldots, N\}$ **do**
3:    Collect $M$ samples using $\pi_k$, add them to replay buffer $\mu$, sample $\{(s_i, a_i)\}_{i=1}^N \sim \mu$.
4:    Evaluate $Q_\theta(s, a)$ and $\kappa_\psi(s, a)$ on samples $(s_i, a_i)$.
5:    Compute target values for $Q$ on samples:
      $y_i = r_i + \gamma \max_{a'} Q_{k-1}(s'_i, a')$.
      $\hat{a}_i = \arg \max_a Q_{k-1}(s'_i, a)$.
6:    Optimize $\kappa_\psi$ using

$$L_\kappa(\psi) := \mathbb{E}_{\mathcal{D}_s} \left[ f^* \left( f' \left( \kappa_\psi(s, a) \right) \right) \right] - \mathbb{E}_{\mathcal{D}_f} \left[ f' \left( \kappa_\psi(s, a) \right) \right].$$

7:    Compute $w_k$ using

$$w_k(s, a) \propto \frac{d^{\pi_k}(s, a)}{\mu(s, a)} \exp \left( - \mathbb{E}_{q_{k-1}(\tau)} \text{TCE}_c(s, a) \right).$$

8:    Minimize Bellman error for $Q_\theta$ weighted by $w_k$.
      $\theta_{k+1} \leftarrow \underset{\theta}{\arg\min} \frac{1}{N} \sum_i^N w_k(s_i, a_i) (Q_\theta(s_i, a_i) - y_i)^2$.
9: **end for**

# D   Experiments

We now present some additional experimental results and experiment details which we could not present due to shortage of space in the main body.

## D.1   Cumulative Recurring Probability on Atari Games

Table 1: The value of $\epsilon_\pi$ with different policies in Atari games.

|              | Initial (Random) policy | Policy at timestep 100k | Policy at timestep 200k |
|--------------|-------------------------|-------------------------|-------------------------|
| Pong         | 0.00                    | 0.00                    | 0.00                    |
| Breakout     | 0.00                    | 0.00                    | 0.00                    |
| Kangaroo     | 0.44                    | 0.32                    | 0.15                    |
| KungFuMaster | 0.66                    | 0.06                    | 0.01                    |
| MsPacman     | 0.44                    | 0.04                    | 0.00                    |
| Qbert        | 0.02                    | 0.05                    | 0.00                    |
| Enduro       | 0.00                    | 0.00                    | 0.00                    |

In Pong, Breakout and Enduro, $\epsilon_\pi$ keeps zero, so there is no error terms in such environments. For KungFuMaster and MsPacman, though $\epsilon_\pi$ is high for the initial policy, its value decreases rapidly as the policy updates. The error term in Kangaroo induces some error but $\epsilon_\pi$ is still much smaller than one. The experiment results imply we can ignore the error term in most reinforcement learning environments.

## D.2   Illustrations on Stable Temporal Structure

We conduct an extra experiment in the GridWorld environment to support our claim that the trajectories have a stable temporal ordering of states. Fig. 1 shows an empirical verification of the stable temporal ordering of states property.

The result shows that the variance of distance to end in one state is not large and decreases fast in training process. This means the property is not a strong assumption and can be satisfied in many environments.

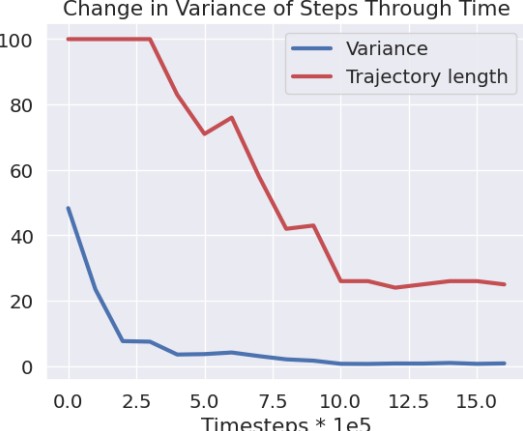

Figure 1: Change in variance of distance to end through time. For each timestep, the red line shows the average trajectory length in the last 500 states. The blue line shows the average variance of the last 500 states, where the variance for each state is calculated from its positions in their corresponding trajectories.

## D.3 Description of Involved Environments

The Meta-World benchmark [5] includes a series of robotic manipulation tasks. These tasks differ from traditional goal-based ones in that the target objects of the robot. For example, the screw in the hammer task has randomized positions and can not be observed by RL agents. Therefore, Meta-World suite can be highly challenging for current state-of-the-art off policy RL algorithms. Visual descriptions for the Meta-World tasks are shown in Fig. 2. DisCor [2] showed preferable performance on some Meta-World tasks compared to SAC and PER [6], but the learning process is slow and unstable.

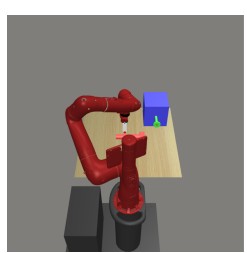 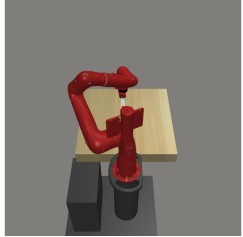 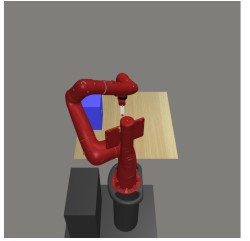 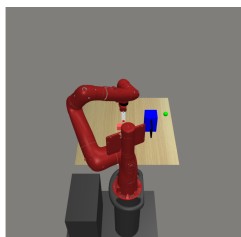

Figure 2: Pictures for Meta-World tasks hammer, sweep, peg-insert-side and stick-push.

## D.4 Extended Results on Atari Environment

We evaluate ReMERN on an extended collection of Atari environments. As is shown in Tab. 2, ReMERN outperforms baseline methods in most of the environments.

## D.5 Extended Evaluation on Gridworld

Aside from the FourRooms environment in Gridworld, we also conduct comparative evaluation on the Maze environment. The results are shown in Fig. 3. The Maze environment perfectly fits for our TCE-based prioritization, and TCE achieves the best performance among other methods.

## D.6 The Relation Between Distance to End and $|Q_k - Q^*|$

In section D.5, the relationship between $|Q_k - Q^*|$ and distance to end has been shown in tabular environments. In this section, we explore the relationship in environments with continuous state and

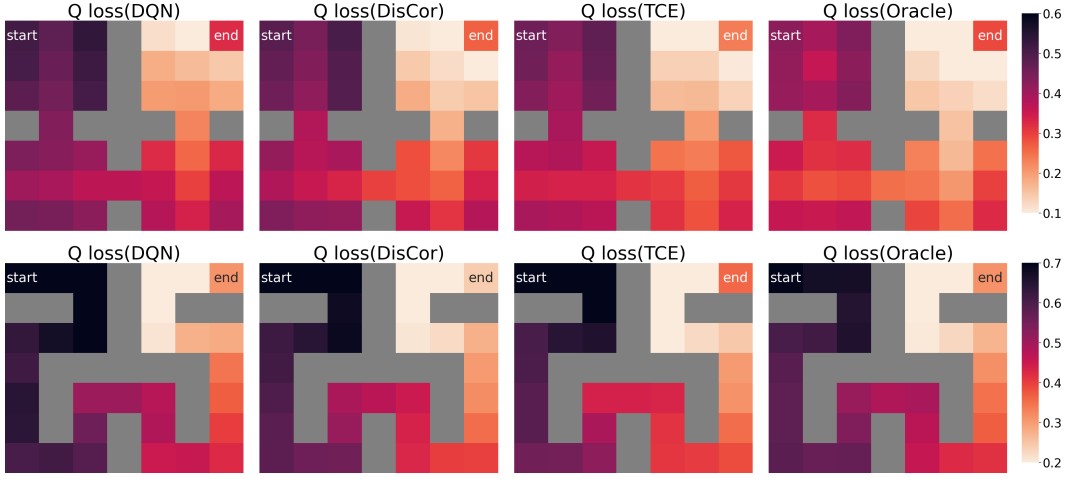

Figure 3: Extended evaluation results on Gridworld.

action spaces, i.e., Ant and Hopper tasks of MuJoCo environment. Since $Q^*$ is inaccessible in these complex continuous control tasks, we approximate it by doing Monte-Carlo rollout using the best policy during training. The results are shown in Fig. 4.

The negative correlation between the two quantities is obvious in Ant-v2, but vague in Hopper-v2. It is because Hopper is a relatively easy task so that all state-action pair have small Q loss and don't have such correlation. The performance of ReMERT shown in Section 4 accords with this observation. ReMERT outperforms other algorithms in environments with a high correlation between the two quantities, and has a relatively poor performance in environments without such correlation.

## D.7 Implementation Details

### D.7.1 Algorithm Details

**Weight Normalization** To stabilize the prioritization, we apply normalization to the estimation of two terms: $\frac{d^{\pi_k}(s,a)}{\mu(s,a)}$ and $\exp(-|Q_k - Q^*|)$.

Table 2: Extended experiments on Atari.

| Environments | DQN(Nature) | DQN(Baseline) | PER(rank-b.) | ReMERT(Ours) |
|---|---|---|---|---|
| Assault | 3395±775 | 8260±2274 | 3081 | **9952**±3249 |
| BankHeist | 429±650 | 1116±34 | 824 | **1166**±82 |
| BeamRider | 6846±1619 | 5410±1178 | **12042** | 5542±1577 |
| Breakout | 401±27 | 242±79 | **481** | 223 ±79 |
| Enduro | 302±25 | 1185±100 | 1266 | **1303**±258 |
| Kangaroo | 6740±2959 | 6210±1007 | **9053** | 7572±1794 |
| KungFuMaster | 23270±5955 | 29147±7280 | 20181 | **35544**±8432 |
| MsPacman | 2311±525 | 3318±647 | 964.7 | **3481**±1350 |
| Riverraid | 8316±1049 | 9609±1293 | 10205 | **10215**±1815 |
| SpaceInvaders | **1976**±893 | 925±371 | 1697 | 877±249 |
| UpNDown | 8456±3162 | 134502±68727 | 16627 | **145235**±94643 |
| Qbert | 10596±3294 | 13437±2537 | 12741 | **14511**±1138 |
| Zaxxon | 4977±1235 | 5070±997 | **5901** | 5738±1296 |

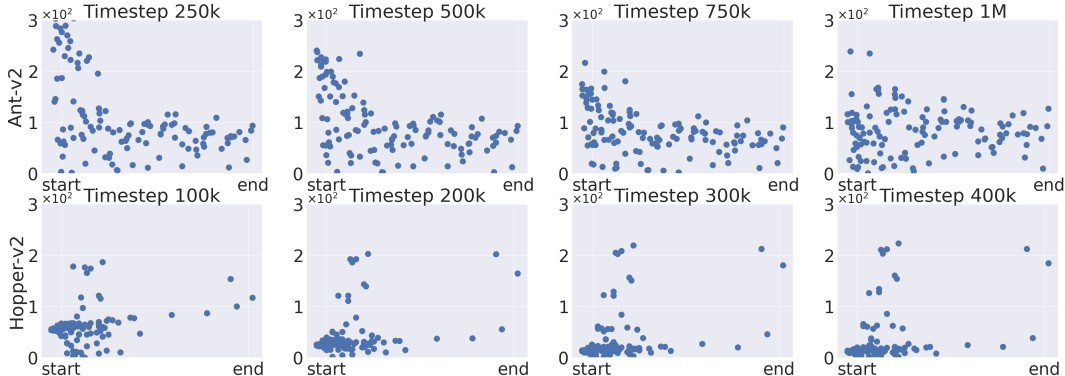

Figure 4: The relationship between $|Q_k - Q^*|$ and distance to end in two MuJoCo tasks (Ant and Hopper).

First, we introduce the normalization in calculating $\frac{d^{\pi_k}(s,a)}{\mu(s,a)}$, which aims to address the finite sample size issue. The normalization is:

$$\tilde{\kappa}_\psi(s,a) := \frac{\kappa_\psi(s,a)^{1/T}}{\mathbb{E}_{\mathcal{D}_s}[\kappa_\psi(s,a)^{1/T}]}$$

where $\mathcal{D}_s$ is the slow buffer and $T$ is temperature.

ReMERN uses $\Delta_\phi$ to fit the discounted cumulative Bellman error. However, the Bellman error has different scales in various environments, leading to erroneous weight. We normalize it by dividing a moving average of Bellman error. The divisor is denoted as $\tau$. Then the estimation of $\exp(-|Q_k - Q^*|)$ becomes

$$\exp\left(-\frac{\gamma\left[P^{\pi^{w_{k-1}}}\Delta_{k-1}\right](s,a)}{\tau}\right)$$

**Truncated TCE**  TCE may suffer from a big deviation when $h_\tau^\pi(s,a)$ is too large or too small. To tackle this issue and improve the stability of the prioritization, we clip the output of TCE into $[b_1, b_2]$, where $b_1$ and $b_2$ are regarded as hyperparameters.

**Baselines**  For the ReMERN and ReMERT algorithms in continuous action spaces with sensory observation, we alter the re-weighting strategy to $\frac{d^{\pi_k}(s,a)}{\mu(s,a)}$ and TCE approximation based on the source code provided by DisCor[1]. For the algorithms in discrete action spaces with pixel observation, we employ the baseline Tianshou[2] [7] and add corresponding components.

### D.7.2  Hyperparameter Details

The hyperparameters of our ReMERN and ReMERT algorithms include network architectures, learning rates, temperatures in on-policy reweight and DisCor, and the lower and upper bound in TCE algorithm. They are specified as follows:

- **Network architectures**  We use standard Q and policy network in MuJoCo benchmark with hidden network sizes [256, 256]. In Meta-World we add an extra layer and the hidden network sizes are [256, 256, 256]. The networks computing $\Delta$ and $\kappa$ have one extra layer than the corresponding Q and policy network.

- **Learning rates**  The learning rate for continuous control tasks, including Meta-World, MuJoCo and DMC, is set to be 3e-4 for Q and policy networks alike. For Atari games, the learning rate is set to be 1e-4 and fixed across all environments.

---

[1]https://github.com/ku2482/discor.pytorch
[2]https://github.com/thu-ml/tianshou

- **Temperatures** The temperature for weights related with $\frac{d^{\pi_k}(s,a)}{\mu(s,a)}$ is 7.5 and fixed across different environments. Also, DisCor has a temperature hyperparameter related to the output normalization of the error network. We keep it unchanged in the Meta-World and DMC benchmark, and divide it by 20 in MuJoCo environments to make it compatible with on-policy prioritization weights.

- **Bounds in TCE** We select time-adaptive lower and upper bounds for TCE. The lower bound rises from 0.4 when training begins to 0.9 when it ends, and the upper bound drops from 1.6 to 1.1 accordingly. The bounds are fixed across different environments.

- **Random Seeds** In MuJoCo, Meta-World and DMC benchmarks, we run each experiment with four random seeds. The results are plotted with the mean of the four experiments. In Atari games, we run experiments with three random seeds and select the one with max return.