# OpenReview forum: "Regret Minimization Experience Replay in Off-Policy Reinforcement Learning"
_NeurIPS.cc/2021/Conference — NeurIPS 2021 Poster_

### Official Review · Reviewer_okGN · 2021-07-16

**Rating:** 6
**Confidence:** 4

**Summary:**

This paper proposes an approach to prioritizing experience replay, specifically it advocates for prioritizing experiences that, when sampled and used for Bellman error minimization, result in lower regret (or really higher reward objective).



**Limitations And Societal Impact:**

Yes

**Main Review:**

The approach taken here to prioritize experience is certainly interesting, as it actually connects the prioritization problem to the reward maximization problem, which is the ultimate goal of RL. As such, the prioritization approach could be thought of as objective-aware. This stands in contrast to previous work in the space that defines a good prioritization scheme as one that, for example, reduces the TD error rapidly (PER).

The paper nicely demonstrates that heuristics, such as the one used by PER, may not necessarily manifest themselves into higher rewards, and so it nicely motivates why we need to reformulate the prioritization problem.

The solution to the reformulated problem highlights key factors when determining which sample to use:
1- high hindsight Bellman error
2- low importance sampling (between behavior and target policies)
3- accuracy of the resultant Q*
4- focusing on actions behind which the policy assigns more probability.

Thoughts regarding the proof of Theorem 1: assumption 1 sounds highly arbitrary and self-serving. Is it correct to say that the theory won't apply to MDPs in which a transition to a same state is likely? It seems like the assumption is only used to approximate the gradient in the follow up derivation. Is it not possible to remove the assumption and bound the error instead?

I didn't get why we used Jensen's inequality in (14). Why can't we directly optimize for 12? In (16), why can't a p_k be strictly zero? In step 4, second equality, should a s be a'? In general, i think the proof needs to get polished and assumption better justified before this result is ready for publication.

The temporal property of Q error seems kind of hacky. a) it depends on the current policy, so it needs to be re-estimated with every change, and b) it is not obvious how to think about it in non-episodic domains and/or domains with frequent self-transitioning states. Can you clarify?

Can you also include the computation time required for REMERN and REMERT? In particular, i am interested to know its comparison with DQN and PER. Is the computation of w the bottleneck of the algorithm, or that it does not affect the overall computation time significantly. Also, I'd move the PER result from the Appendix to the main text, as PER would be the most important baseline to assess the competency of the proposed algorithms.

In general, this paper is identifying a weakness of the previous work pertaining to experience prioritization, which is really nice, but i have my own doubts regarding the proposed way of computing the weight. Overall, I lean towards acceptance at this point.

**Time Spent Reviewing:**

4

---

> ### Author Response · Authors · 2021-08-10
> **Authors' Response**
>
>
> Thank you for your thoughtful and inspiring comments. We provide discussions and explanations about your concerns as follows.
>
> **Q: Assumption 1 sounds highly arbitrary and self-serving. Is it correct to say that the theory won't apply to MDPs in which a transition to a same state is likely? It seems like the assumption is only used to approximate the gradient in the follow up derivation. Is it not possible to remove the assumption and bound the error instead?**
>
> A: First, it makes sense to remove the assumption and bound the error instead. We will change our theorem to an assumption-free form in the revised version of this paper. In this way, we can apply our theorem to more general MDPs. Second, our assumption is related to the traditional recurrency property. The recurrent time focuses on the expectation time of coming back to a state, while our assumption focuses on the probability of coming back to a state in a certain time period.
>
>
>
> **Q: Why do you use Jensen's inequality in (14)? Why can't you directly optimize for 12?**
>
> A: If we remove Jensen's inequality, the optimization problem can not satisfy LICQ, so that we can not use KKT conditions.
> LICQ is the short for Linear Independence Constraint Qualification, meaning that the set of active constraint gradients $\\{\nabla h_{1}\left(x^{\ast}\right), \ldots, \nabla h_{m}\left(x^{\ast}\right), \nabla g_{i}\left(x^{\ast}\right)\\}$ is linearly independent. In the case of optimization without Jensen's inequality, we have two sets of active constraints, namely $w_k(s,a)\geq 0$ and $\mathbb E_{\mu(s,a)}w_k(s,a)=1$. They will have linearly dependent gradients, so we can not use KKT conditions. In contrast, if Jensen's inequality is involved in the proof, the constraints $w_k(s,a)\geq 0$ won't be active, and the KKT conditions can be used to compute the optimal solution.
>
>
>
> **Q: In (16), why can't a $p_k$ be strictly zero? In step 4, second equality, should a be a'?**
>
> A: (a). $p_k$ can be strictly zero. If $p_k=0$,  we have $d^{\pi_k}(s,a)=0$ or $|Q_k-\mathcal{B}^*Q_{k-1}|(s,a)=0$ by its expression. In such circumstance, the value of $\mu$ doesn't influence the value of $p_k$. Therefore, we can always let $\mu=0$.
> (b). $a$ should be $a'$.
>
> We will add these corrections in the revised version of our paper.
>
> **Q: The temporal property of Q error seems kind of hacky. a) it depends on the current policy, so it needs to be re-estimated with every change, and b) it is not obvious how to think about it in non-episodic domains and/or domains with  frequent self-transitioning states. Can you clarify?**
>
> A: a) We need to obtain the expectation $\mathbb{E}\_{\tau}[h_\tau^{\pi_{k-1}}(s,a)]$ to leverage the temporal property of Q error, where $h_\tau^{\pi_{k-1}}(s,a)$ is the distance of $(s,a)$ to the terminal state in the trajectory $\tau$. Technically we do need to re-estimate $h_\tau^{\pi_{k-1}}(s,a)$ with every policy change. But in practice we find that using $h_\tau(s,a)$ from an outdated policy (e.g. the policy that collects $(s,a)$) can also provide decent information about the distance of $(s,a)$ to the terminal state. For example, in the Humanoid environment the state that the agent is going to fall inevitably will have small $h$ regardless of the current policy. Therefore, we do not involve the additional rollout in our practical algorithm.
>
> b)   ReMERT might not provide proper prioritization weights in non-episodic or frequent self-transitioning tasks since the "Distance to end" or $h(s,a)$ can be ill-defined. In such tasks, our ReMERN algorithm is a better choice.
>
>
> **Q: Can you include the computation time required for ReMERN and ReMERT? In particular, I am interested to know its comparison with DQN and PER. Is the computation of w the bottleneck of the algorithm, or that it does not affect the overall computation time significantly.**
>
>
> A: Comparative experiments on Atari with DQN and PER are time-consuming and we here provide experiments on MuJoCo instead. Comparisons of the computation time required for SAC, DisCor, PER, ReMERN and ReMERT are shown in the following table. Experiments are conducted with one NVIDIA 2080Ti GPU on the Ant-2d environments.
>
> |         Algorithm        |  SAC  |  PER  | DisCor | ReMERN | ReMERT |
> | --------- | --------- | ----------- | ----------- | ----------- | ------------ |
> |    Time/h for 1M step    |  4.88 |  6.73  |  6.87  |  8.13  |  6.45  |
> |Time/h to reach reward 5k |  4.92 |  23.21   | 10.98 |  13.66 |  4.51  |
>
>
> As shown by the results, prioritization methods like DisCor, ReMERN and ReMERT will indeed increase the time cost for each training step due to the neural networks, but these extra methods will not become the bottleneck of the computation time. Also, ReMERT can be more time efficient than SAC to reach a reasonable performance threshold.

---

> > ### Comment · Reviewer_okGN · 2021-08-15
> > **Assumption 1**
> >
> > Sorry to linger on this, but the explanation regarding assumption 1 did not make sense. Can I have a pointer to another work that has made the same or at least a similar assumption?
> >
> > Can I at least get a sketch of how the proof will work in light of removing the assumption and bounding the error?
> >
> > Thanks for the explanation regarding the application of Jensen.

---

> > > ### Author Response · Authors · 2021-08-18
> > > **Authors' Response**
> > >
> > > 1. We are unable to find another work with the same assumption, but a similar statement is common in Stochastic Process. For a given policy $\pi$, the state sequence $(s_t)_{t=0, 1, 2, \dots}$ can be viewed as a stochastic process. Using its terminology, the notation $\rho^\pi(s,a,t)$ is the *hitting probability* of state $s$ at time step $t$.
> > >
> > >     Previous researches [Avrachenkov et al., 2018][Bean et al., 2005][Palacios et al., 2014][Stadje, 1987] on this problem mainly focus on calculating the hitting probability or corresponded hitting time in various stochastic processes. Based on their results, we find most of the stochastic processes satisfy Assumption 1 when parameters are properly chosen. For example, consider an asymmetric random walk [Avrachenkov et al., 2018][Stadje, 1987] on a two dimensional lattice, let the probability of going left be 0.9, and the probability of going to other three direction be 0.1/3 respectively. The probability is assigned to mimic the $\epsilon$ greedy policy in reinforcement learning, where $\epsilon$ is usually 0.1. In this setting, $\sum_{t=1}^H\rho^\pi(s,a, t)\leq 0.09$, which is a small number and the assumption is satisfied. In fact, Assumption 1 is more likely to hold in more complicated environments, such as the common tasks that Reinforcement Learning aims at solving. We have conducted experiments to verify this, the results of which can be found in Tab.1 in https://anonymous.4open.science/r/ReMERN-T-C6B4.
> > >
> > > 2. You can find the proof at https://anonymous.4open.science/r/ReMERN-T-C6B4.
> > >
> > >
> > > **Reference**
> > >
> > > [Avrachenkov et al., 2018] Avrachenkov, K., Piunovskiy, A., and Zhang, Y. (2018). Hitting times in markov chains with restart and their application to network centrality. Methodology and Computing
> > > in Applied Probability, 20(4):1173–1188.
> > >
> > > [Bean et al., 2005] Bean, N. G., O’Reilly, M. M., and Taylor, P. G. (2005). Hitting probabilities and
> > > hitting times for stochastic fluid flows. Stochastic processes and their applications, 115(9):1530–1556.
> > >
> > > [Palacios et al., 2014] Palacios, J. L., Gómez, E., and Del Río, M. (2014). Hitting times of walks on
> > > graphs through voltages. Journal of Probability, 2014.
> > >
> > > [Stadje, 1987] Stadje, W. (1987). The exact probability distribution of a two-dimensional random walk. Journal of statistical physics, 46(1):207–216.

---

### Official Review · Reviewer_hPUt · 2021-07-16

**Rating:** 3
**Confidence:** 4

**Summary:**

This paper attempts to provide a theoretical approach to the prioritization of the experience reply to minimize regret.
The main result of the paper shows that to minimize the regret, one should minimize a weighted mean squared Bellman error, where the weights $\omega(s, a)$ depend on four terms

1. the magnitude of the Bellman error for the state-action pair $s,a$
2. the degree of "policiness" of the sample $s, a$ (seen as the probability of sampling from the experience replay v.s. the discounted state distribution)
3. the magnitude of the error $|Q_k - Q^*|(s, a)$, which tells us how "wrong" is $Q_k(s, a)$
4. the probability of sampling $a$ given $s$ according to the current policy.

The authors develop some heuristics to estimate these four components, and to weigh the replay memory accordingly.



**Limitations And Societal Impact:**

Although the author claimed that they discussed the limitations of their work, I do not find the discussion in the paper.
I suggest that the discussion should be made more clear in the Conclusion.

There is no societal impact to be discussed.

**Main Review:**

The idea to find a principled way to prioritize the replay buffer is nice.
Theorem 1 (if correct) is an important contribution, as it allows to use the experience replay in a smarter way.

I have, however, a few fundamental concerns. The first one is about why the authors focus on the regret. In their setup, the maximization of the return should be enough, and I find the terminology "regret", confusing, as it is used usually in a different context.
Another concern is that the paper looks like more a draft stage rather than a finished work. There are a few imprecisions, as well as many details missing. I list them later, but to give an idea, many mathematical symbols are not defined, and some of the reasonings made do not make perfect sense to me.
My last concern regards the amount of approximations introduced. It is not easy (at least for me) to understand the logic behind the approximations. Unfortunately, the language used in the paper does not help to clarify them.



The authors evaluate their methods (ReMERN and ReMERT) on different settings. I think that the empirical section is well constructed (even though I would advocate the comparison with the classic Prioritized Experience Replay); however, I do not find the results to be particularly significant, especially when taking in account that the method proposed is strongly empirical (I usually tend to weight less the importance of the experimental section when the theoretical section is stronger).


__Questions__

1. In lines $\texttt{141-144}$ you state that "an efficient update requires $\omega_k$ to be on-policy". I do not understand this sentence. If I understand Theorem~1 correctly, when $d^{\pi, k}(s, a)$ is low and $\mu(s, a)$ is high, then $\omega_k(s, a)$ will be low, while when $d^{\pi, k}(s, a)$ is high and $\mu(s, a)$ is low, $\omega_k(s, a)$ will be high. How is this related to on-policy/off-policy? It seems that in the two cases that I mention, clearly both off-policy, we have two different asymptotic $\omega_k(s, a)$.

2. I do not understand Equation 4. I do not know what is a slow replay buffer and a fast one, and I do not know the meaning of $f^*$ and $f'$

3. How do you train $\Delta_k$? From the formula seems that you need to store all the previous policies $\pi_k$ (small detail: you usually do not have $P^{k-1}$ or $\mathcal{B}^*$, I guess that you are fitting a _sample estimate_).

4. Is Equation~  an upperbound, a lower bound, or ... ? It is hard for me to relate it to Equations 2 and 3. (As I discuss later, I am convinced that using $\propto$ is wrong in Eq. 7).

5. $\texttt{247}$ what does it mean "the term $h^\pi_\tau(s, a)$ is relatively fixed"? Means that it has low variance? And what is the _temporal ordering_ of the estates (or a _stable temporary order of states_)?


__"Minor" Questions__

1. I fail to understand the need of minimizing $\eta(\pi*) - \eta(\pi_k)$, since it is equivalent to $\max \eta(\pi_k)$ (up to an additive constant factor).

2. I fail to understand what $|Q_k - Q^*|(s, a)$ means. Is it a shorthand for $|Q_k(s, a) - Q^*(s, a)|$?

3. The policy is defined as $\pi(s) = \mathrm{arg}\max_a Q_k(s, a)$ (line $\texttt{126}$). What is $\pi_k(a|s)$ in Theorem~1? (remember, the paper should be self contained, I should not go to investigate that in the Appendix.)

4. What are $f'$ and $f^*$ in Equation 4?

5. What does it mean in Theorem 2 "Ignoring some terms", "holds with high probability"? I know you are referring to the Appendix, but the paper should be self contained. Including a "Theorem" in such unclear way, is completely useless.


__Side notes__


1. You define $\eta(\pi)  = \mathbb{E}\left[\sum_{t>0}\gamma^t r(s_t, a_t)\right]$  on line $\texttt{62}$, but then you define it as $\eta(\pi) = \mathbb{E}_{d^\pi(s, a)}[r(s,a)]$., where $d^\pi$ is the _discounted stationary state distribution_.
The two definitions are not equivalent as,

$$ (1-\gamma)\mathbb{E}\left[\sum_{t>0}\gamma^t r(s_t, a_t)\right] = \mathbb{E}_{d^\pi(s, a)}[r(s,a)]$$

2. Eq two should be modified in the following way:

$$\text{s.t.} \quad Q_k  = \mathrm{arg}\min_Q E_\mu \left[\omega_k(s, a)  \cdot (Q-\mathcal{B}^*Q_{k-1})(s, a)\right].$$

3. Equation 7 is an approximation of Equation 3; you should not use the same symbol both for the "correct" value and for its approximation.

4. I think at line $\texttt{233}$ $Q$ should be substituted with $Q_{k-1}$.

5. In the Appendix you define $d^\pi$. I find the definition wrong, or at least unclear. Why don't you use the definition from Sutton et al., NIPS 1999?

6. $\texttt{192}$, The first term is the _projection error_.

7. Line $\texttt{233}$. How do you update $Q_k = (1-\alpha)Q_{k-1} + \alpha Q_{k-1}$? Is $Q$ tabular? What it is done usually, it is to update the parameters of $Q_k$ with a soft-update rule.

8. Line $\texttt{236}$. How do you ensure that a small change in the Q-function results in a small change in the policy?


UPDATE
=======

I thank the authors for their answers to my questions.
Unfortunately, after carefully considering the author's response and other reviewer's responses, I am willing to keep my score as it is.

The paper analyzes an interesting problem and proposes a theoretical approach to minimize the regret in weighting the samples coming from the replay memory. The algorithmic solution, however, relies heavily on approximations (like the result in Equation 7, and the need to approximate the Bellman error with a neural network) and heuristics, as I also remarked in my review.
I usually expect strong empirical evidence supporting empirical works. However, in my perspective, the results are not so strong.
In Figure 4, the learning curve is clearly unstable both in push and sweep. In stick-pull, hammer, and peg-insert-side, the variance is considerably high, making me doubtful of the statistical significance of the results. The same issue of high variance seems to be present also in most of the tasks in Figure 3 and Table 1.

As I remarked in the review, I also detected several imprecisions (mathematical notation and language), which make me think that the paper should be polished.

Side remark. The policy is greedy w.r.t. the Q-function, meaning that small changes in the Q-function many still cause large changes in the policy, i.e. consider $Q(s_1, a_1) = 100$ and $Q(s_1, a_2) = 99.99$, which leads a policy $\pi(a_1| s_1) = 1; \pi(a_2|s_1) = 0$, but after an update, we suppose that $Q(s_1, a_1) =100$ and $Q(s_1, a_2) = 100.1$. This will lead the policy to have a big "jump", since now the greedy policy will be $\pi(a_1| s_1) = 0; \pi(a_2|s_1) = 1$.




**Time Spent Reviewing:**

5

---

> ### Author Response · Authors · 2021-08-10
> **Authors' Response**
>
> We thank the reviewer for the thoughtful review and detailed comments. We first provide explanations on your concerns and then answer your questions as follows.
>
> **Q: Why do you introduce the term "regret" since "return maximization" seems to be enough?**
>
> A: The  objective of regret minimization can be directly analysed and approximated by our theoretical analysis. For example, we can derive the distance between $Q_k$ and $Q^*$ in lemma 4 more naturally. Moreover, in this paper we only study the instantaneous regret, while the cumulative regret is also an important performance measure in RL, which may be studied in future work. Introducing the term "regret" in this paper facilitates further researches.
>
>
> **Q: What are the meanings of $f^\ast$, $f'$, $|Q_k-Q^\ast|(s,a)$ and $\pi_k(a|s)$ respectively?**
>
> A: $f^*$ is the convex conjugate of $f$ and $f'$ is the derivative of $f$, $|Q_k-Q^*|(s,a)$ is the shorthand for $|Q_k(s,a)-Q^*(s,a)|$, and $\pi(a|s)$ is the probability that choose $a$ under state $s$. we will (1) add the definition of $f^*$, $f'$, $|Q_k-Q^*|(s,a)$ and $\pi_k(a|s)$, (2) remove improper usage of $\propto$, (3) modify the definition of $d^\pi(s,a)$ to the standard definition, (4) use projection error instead of optimization error. Thanks for your suggestions on improving the clarity of paper.
>
> **Q: Are all the approximations made in this paper necessary? Can you provide some rationalization or intuition for these approximations?**
>
> A: Approximations are necessary for solving the optimization problem in Section 3.2 and proposing practical algorithms. It is extremely difficult, if not impossible, to deal with all complex RL problems with various kinds of environments, dynamics and rewards by solving one optimization problem. We have to ignore some factors irrelevant to prioritization weights, or
> quantities which are computationally expensive to obtain during training.
>
>
> **Q: Why do not you compare with PER? Is the improvement significant against baseline methods?**
>
> A: [Kumar et al., 2020] and [Sinha et al., 2020] have already shown in their experiments that PER performs worse than SAC in continuous control tasks like MuJoCo and Meta World, so we do not include PER as a baseline in these tasks. We do include PER as baseline in Atari benchmarks and show the results in Appendix D.
>
> Regarding the significance of improvement, previous prioritization methods like PER and DisCor struggle to outperform SAC in the MuJoCo benchmark as mentioned before, but  we have made reasonable empirical improvements in MuJoCo compared with these methods. Experiments on Meta-World and Atari are even more significant than ones on MuJoCo.
>
>
> **Q:  How is $\frac{d^{\pi_k}(s,a)}{\mu(s,a)}$ related to on-policy / off-policy data distribution?**
>
> A:  $\mu$ is the distribution of the state-action pairs in the replay buffer, which is an off-policy distribution agnostic to specific policies. $d^{\pi_k}$ is the distribution induced by the current policy $\pi_k$, which is an on-policy distribution. So $\frac{d^{\pi_k}(s,a)}{\mu(s,a)}$ serves as the importance weight between the on-policy and off-policy data. The expectation with respect to the off-policy data distribution $\mu(s,a)$ multiplied by the importance weight is equal to the expectation with respect to the on-policy data distribution $d^{\pi_k}(s,a)$. Our prioritization weight $w_k(s,a)$ in Eq. (2) contains such an importance sampling term, so it will prefer samples from the on-policy distribution.
>
>
> **Q: What is a slow replay buffer and a fast one?**
>
> A: The slow buffer is the traditional replay buffer in off-policy learning containing data from distribution $\mu$, while the fast buffer is many times (usually 10x) smaller than the slow buffer and contains only a small set of trajectories from very recent policies. These trajectories tend to be closer to the on-policy distribution $d^{\pi_k}$ compared with samples from the slow buffer.
>
>
> **Q: How to train $\Delta_k$?**
>
> A: Eq. (6) shows the update rule of $\Delta_k$. It is similar to Bellman equation if $|Q_k-\mathcal{B}^*Q_{k-1}|$ was substituted with the reward. So we can use neural networks to represent $\Delta_k$, just as the Q network in the Bellman equation. The training of $\Delta_k$ is also the same as deep Q learning, with transitions sampled from the replay buffer. The source code will also be released soon for closer inspections.
>
>
> **Q: How is the bounds related to equation 2 and 3?**
>
> A: We use these bounds to derive a lower bound of $w_k$, so that we may down-weight some transitions but never up-weight a transition by mistake. Concretely, $|Q_{k-1}-\mathcal{B}^\ast Q_{k-2}|\leq c_2$ implies $\gamma P^{\pi_{k-1}}\Delta_{k-1}+c_2$ is an upper bound of $|Q_k-Q^*|$. Together with $2-\pi_k(a|s)\geq 1$ and $|Q_{k-1}-\mathcal{B}^*Q_{k-2}|\geq c_1$ we have equation 7 as a lower bound of $w_k$ by omitting the constants.
>
>
> **Q: What is the meaning of $h^\pi_\tau(s,a)$ being relatively fixed? What is the temporal ordering of the states (or stable temporal ordering of the states)?**
>
> A: "$h^\pi_\tau(s,a)$ being relatively fixed" means the variance $\mathrm{Var}_\pi [h^\pi_\tau(s,a)]$ is small. The temporal ordering of the states is the order of states being visited in a trajectory. Stable temporal ordering of the states is another statement that the variance $\mathrm{Var}_\pi [h^\pi_\tau(s,a)]$ is small. We rewrite the latter part of Section 3.4.2 to make it more clear in the revised version of our paper.
>
>
> **Q: What does it mean in Theorem 2 "Ignoring some terms", "holds with high probability"?**
>
> A: We ignore the term $g(\delta', k)$, which equals to $\sqrt{2f(h^\pi_\tau(s,a))^2\phi_k^2 \log \frac{2}{\delta'}}$ when $h_\tau^\pi(s,a)<H$,  and  $\sqrt{2(f(h_\tau^{\pi_k})+\frac{\gamma^{H+1}}{1-\gamma})^2\phi_k^2 \log \frac{2}{\delta'}}$ when $h_\tau^\pi(s,a)\geq H$, where $\phi_k=O(\nu^k)$ and $\nu<1$. This term converges to zero exponentially. "with high probability" means with probability at least $1-\delta$, where $\delta$ is a small number related to $\delta'$ as defined previously. We will make the statement more clear in the revised version of our paper.
>
>
> **Q: How do you update $Q_k=(1-\alpha)Q_{k-1}+\alpha(\mathcal{B}^*Q_{k-1}-Q_{k-1})$? Is  $Q$ tabular?**
>
> A: No, $Q_k$ is represented by a neural network. In such a case, the update is done by gradient descent with a small learning rate. $Q_k$ can be regarded as a small step update from $Q_{k-1}$.
>
>
> **Q: How do you ensure that a small change in the Q-function results in a small change in the policy?**
>
> A: Since the policy is derived from Q function, a small change in Q often leads to a small change in policy. Using previous Q value to approximate the current Q is a commonly employed approximation.
>
>
> **Reference**
>
> [Kumar et al., 2020] Kumar, A., Gupta, A., and Levine, S. (2020). Discor: Corrective feedback in reinforcement learning via distribution correction.
>
> [Sinha et al., 2020] Sinha, S., Song, J., Garg, A., and Ermon, S. (2020). Experience replay with likelihood-free importance weights.

---

> > ### Comment · Reviewer_hPUt · 2021-08-10
> > **A brief answer**
> >
> > 1.  The problem
> >
> > $$\min \eta(\pi^*) - \eta(\pi_k)$$
> >
> > is equivalent to
> >
> > $$\max  \eta(\pi_k)$$
> >
> > since $\eta(\pi^*)$ is.constant. That is why I am saying that speaking about regret, in my understanding, does not really make sense. May I have a precise answer for this concern?
> >
> > 2. With regard to the importance sampling term. I do understand that $d^\pi/\mu$ is an importance sampling correction. What I meant is that it can be both low and high when data is off-policy. For example, if we have $d^\pi(s)$ high and $\mu(s)$ low (therefore sample $s$ is off-policy), then $d^\pi(s)/\mu(s)$ will be high. When, on contrary, $d^\pi(s)$ is low and $\mu(s)$ is high, the sample $s$ is still off-policy, but the ratio will have a low value. Therefore, the importance sampling is not a good indicator of the "off-policy" of the data. Could you clarify?
> >
> > 3. I would clean the "functional notation" $Q_k = (1-\alpha)Q_{k-1} + \alpha(\mathcal{B}^*Q_{k-1} - Q_{k-1})$, as with functional approximation it is not usually possible to update the Q function in that way. there is a projection that must be considered.
> > you could introduce a parameter $\omega_k$ and
> >
> > $$\omega_k = \omega_{k-1} - \frac{\alpha}{2} \nabla_\omega \mathbb{E}\left[\left(Q_\omega(s, a) - (\mathcal{B}^*Q_{k-1})(s, a)\right)^2\right]$$
> >
> > $$\implies \omega_k  = \omega_{k-1} - \frac{\alpha}{2} \mathbb{E}\left[\left(Q_\omega(s, a) - (\mathcal{B}^*Q_{k-1})(s, a)\right)\nabla_\omega Q_\omega(s, a)\right]$$
> >
> >
> > Another possibility would be to say something like
> >
> > $$Q_k = \mathrm{arg}\min_{Q \in \mathcal{Q}} \mathbb{E}\left[\left(Q_\omega(s, a) - (\mathcal{B}^*Q_{k-1})(s, a)\right)^2\right]$$
> >
> > where $\mathcal{Q}$ is the functional space of your $Q$-functions.
> >
> > ---
> >
> > Thanks for your time!

---

> > > ### Author Response · Authors · 2021-08-10
> > > **Authors' Response**
> > >
> > > Thank you for your comments. We provide explanations to three of your concerns.
> > >
> > > 1. The objective of regret minimization makes sense in that it indeed helps simplify the proof. There is a lemma stating that
> > > $\eta(\pi^{\ast})-\eta(\pi_k)=-\frac{1}{1-\gamma}\mathbb{E}\_{d^{\pi_k}(s,a)}A_{\pi^*}(s,a)$, where $A_{\pi^*}(s,a)$ is the advantage function.
> > > We can derive this directly from the regret minimization objective.
> > >
> > >     The choice of regret is also due to a preference of terminology. In fact, "regret" is widely used in RL literature, e.g., [Azar et al., 2017][Jin et al., 2020][Ayoub et al., 2020].
> > >
> > > 2. We may have different understandings about when a sample $s$ is on-policy. In our paper, a sample $s$ being on-policy means $d^\pi(s)>\mu(s)$. So if $d^\pi(s)$ is high and $\mu(s)$ is low, $s$ is a on-policy sample. As to the importance weight $\frac{d^\pi(s,a)}{\mu(s,a)}$, it will be greater than one if $(s,a)$ is a on-policy sample, thus being preferable to the sample. We are sorry for the unspecified statements and extra explanations will be added to the revised version of our paper.
> > >
> > >
> > > 3. Your notation is more clear to clarify the update of $Q_k$ and $w_k$. We will revise these notations in our paper. Thank you for your detailed comments.
> > >
> > > **References**
> > >
> > > [Azar et al., 2017] Azar, M. G., Osband, I., and Munos, R. (2017). Minimax regret bounds for reinforcement learning.
> > >
> > > [Jin et al., 2020] Jin, C., Yang, Z., Wang, Z., and Jordan, M. I. (2020).  Provably efficient reinforcement learning with linear function approximation.
> > >
> > > [Ayoub et al., 2020] Ayoub,  A.,  Jia,  Z.,  Szepesv ́ari,  C.,  Wang,  M.,  and  Yang,  L.  (2020).   Model-based reinforcement learning with value-targeted regression.

---

> > > > ### Comment · Reviewer_hPUt · 2021-08-13
> > > > **Thanks**
> > > >
> > > > I see that you are correct on Point 2. My reasoning was simply wrong.

---

> > > > > ### Comment · Reviewer_Giki · 2021-08-29
> > > > > **makes sense**
> > > > >
> > > > > favouring on-policy samples in replay makes sense

---

### Official Review · Reviewer_Giki · 2021-07-18

**Rating:** 9
**Confidence:** 5

**Summary:**

Regret
This paper explores a theoretical understanding of a current very prevalent method of training deep RL agents: experience replay. It brings an optimization perspective and formulate return maximization using an importance weighting to samples in buffer. The most important contribution of this paper is coming with a solution of weight that is almost analytical, which explains quite a few interesting myth/questions regarding experience replay. (Detailed below).


**Ethics Review Area:**

["I don’t know"]

**Main Review:**


Figure 1: the x axis is iteration? What’s the explanation that PER and Discor performing worse than uniform? Any influence from experiment parameters? How many runs are here? Step-size? Number of samples? What are the features? Initialization? What is the particularity of this problem that makes the two methods slower? It is foreseeable that these setup can influence the performance on such a simple example and such a short time of training. I see you have some explanations from line 99, but still it’s not very convincing. Looks like more like an argument than a solid explanation.

This doesn’t seem a good example (at least it is not well presented: not convincing; no maths or understanding why the two methods are slower than uniform sampling).

Equation 1: this formulation is interesting. Just the dependence of the policy isn’t explicitly dependent on w_k, which is flawed. Perhaps you should define \pi_k from Q_k.

I assume all w_k>=0? Then shouldn’t it be constrained? It could also be in the form of exp^w. In the paper, w is directly learn for each sample. In practice, there are millions to billions of samples,  generalization between samples is important. Thus considering parameterizing w_k will be interesting.

That being said, the experiments missed a study of the weights. Are they negative or positive? Any insights into the signs of w on the performance and behavior of the algorithm?

It looks like an adaptive importance sampling strategy, especially eq 2. In Th 1. Has an importance sampling ratio. I think this is not accidental. I like the “on”-policiness, it is a very interesting understanding of the off-policy samples. The downside is that the learning of w will have a high variance.

Note if Q_k converges to Q at a linear rate (for value iteration, a gamma rate, seems in your eq. 7 it is something like this), then the weights will be exponentially increasing. This will the sampling focus on those samples that will drive Q_k close to Q*. This will be very interesting to show for some generalization case to illustrate this perspective.

The hindsight Bellman Error is interesting too. That it explains hindsight experience replay is most interesting I think.

Action selection perspective: If a_k is selected with a high probability, then the weight for s, a_k is small. Sampling this state action pair will be small in experience replay. So the weighting will prefer those rarely taken actions, correct? If this is correct, I think the last item of your understanding from Th 1, “a_k is less likely to be selected by the policy“, should be stated more precise like, “samples of rarely taken actions will be replayed more”. That “a_k is less likely to be selected by the policy“ is not necessarily true: it may be influenced by features and many other things.

Could you double check the “2-\pi_k” and why it’s not “1-\pi_k”?

By “informal” you mean a relaxation?

Experiments were performed on Mujoco continuous action tasks, and outperforms Discor, and SAC on four tasks (with a closer gap in Walker2d), which is not bad.

Also on Meta-World benchmark (8 tasks were selected). The results are more promising than Mujoco, with larger winning gaps.

The valuation is also done for discrete-action tasks on Atari games. On 5 selected games the algorithm outperforms DQN by a large margin. However, the baseline DQN is weak; you probably needs to include more baselines especially those experience replay,  some model-based methods as well as exploration method (since your weighting relates to action selection perspective).

related: experience replay and Atari baselines

Recurrent Experience Replay in Distributed Reinforcement Learning
https://openreview.net/forum?id=r1lyTjAqYX

Distributional Reinforcement Learning for Efficient Exploration
https://arxiv.org/abs/1905.06125















**Time Spent Reviewing:**

1.2

---

> ### Author Response · Authors · 2021-08-10
> **Authors' Response**
>
> Thank you for your thoughtful comments and inspiring suggestions. We appreciate it that you go along with several of our arguments and acknowledge many of our contributions. We provide explanations to your concerns as follows.
>
> **Q: The motivating example and Figure 1 doesn’t seem a good example (at least it is not well presented: not convincing; no maths or understanding why the two methods are slower than uniform sampling), as well as a few questions about the details of the example.**
>
> A: The motivating MDP in section 3.1 is specially designed to show the suboptimality of previous heuristics for prioritization. Specifically, the estimated Q value are stored in a table and prioritized Q learning is used to update the value estimations of each state-action pair, as described by the equation $Q_k(s,a)=Q_{k-1}(s,a)+\alpha w(s,a)[r(s,a) + \gamma\mathbb{E}\_{s'}\max_{a'}Q_{k-1}(s', a') - Q_{k-1}(s,a)]$, where $\alpha$ is the learning rate and $w$ is the prioritization weight. However, with unconstrained learning, accurate Q values can be obtained in few iterations, which is not the case in reality with function approximations. To simulate the approximation error in complex environments, we initialize all Q table entries with $0$ and set the learning rate $\alpha$ (or step size) as 0.1. The reward scale is set to be around 1. The Q value of certain state-action pairs can be more aggressively updated by assigning larger prioritization weight.
>
> In the MDP described in Figure 1, PER assigns more priority to the "left" action in all states with larger TD error, resulting in a more accurate value estimation on those actions. However, "left" actions are suboptimal on $s_0$, $s_1$ and $s_2$. Therefore, the extra accuracy from PER can not help find the optimal action in this MDP. The same problem exists in DisCor. "Left" actions leading to terminal states have larger "corrective feedback", which are favoured by DisCor.
>
> The experiment results of PER, DisCor and no prioritization (Uniform) are shown in (b)(c) of Figure 1. The x axis represents the number of iterations of Q learning. Due to the problematic heuristics, PER and DisCor are slower in finding the optimal policy than Q learning with no prioritization.
>
>
> **Q: Equation 1: this formulation is interesting. Just the dependence of the policy isn't explicitly dependent on $w_k$, which is flawed. Perhaps you should define $\pi_k$ from $Q_k$.**
>
> A: Our definition of $\pi_k$ follows Eq.1: $\pi_k(s)=\arg\max_a Q(s,a)$, so $\pi_k$ is already dependent on $Q_k$. Probably you mean $\pi_k$ should be defined with $w_k$? This is indeed the case and we will change the notation $\pi_k$ to $\pi_k^{w_k}$ and $Q_k$ to $Q_k^{w_k}$ to emphasize the dependence of $\pi_k$ and $Q_k$ on $w_k$.
>
>
> **Q: I assume all $w_k\geq 0$? Then shouldn't it be constrained? The experiments missed a study of the weights.**
>
> A:  In the proof of Theorem 1, we define $p_k(s,a)$ to be $w_k(s,a)\mu(s,a)$, and $p_k$ is the distribution from which we sample in Q learning. We constrain $p_k\geq 0$ there, and $w_k(s,a)=\frac{p_k(s,a)}{\mu(s,a)}\geq 0$. This is indeed one of the constraints to the optimization problem defined in Section 3.2. We will add it in the revised version of our paper.
>
>
> **Q: In practice, there are millions to billions of samples, generalization between samples is important. Thus considering parameterizing $w_k$ will be interesting.**
>
> A: The idea of parameterizing $w_k$ sounds very interesting. The parameters may be updated with gradient descent from policy or value gradients if we manage to find a differentiable connection between the prioritization weights and the reward.
>
>
> **Q: I think the last item of your understanding from Th 1, "$a_k$ is less likely to be selected by the policy", should be stated more precise like, "samples of rarely taken actions will be replayed more". That "$a_k$ is less likely to be selected by the policy" is not necessarily true: it may be influenced by features and many other things.**
>
> A: We agree with the suggestion. Your statement is more accurate in that the prioritized sampling are performed on transitions already sampled and stored in the buffer.
>
>
> **Q: Could you double check the "$2-\pi_k$'' and why it's not "$1-\pi_k$''?**
>
> A: We have double checked our deriviation, and the term $2-\pi_k$ is exactly part of the solution to the optimization problem in Section 3.2. An intuition why the weight $1-\pi_k$ does not make sense is that suppose our policy decides to choose action $a$ at state $s$ with high probability, then $1-\pi_k$ is close to zero, and this state-action pair is hardly ever sampled. Intuitively, it is not a good strategy.
>
>
> **Q: By "informal" you mean a relaxation?**
>
> A: Informal means we omit some tedious terms and assumptions in the theorem. The formal version is provided in the appendix. For example, we omit an assumption in theorem 1, which is about the recurrent probability of the current policy. For theorem 2, we omit some terms that converge to zero.
>
>
> **Q: The baseline DQN is weak; you probably need to include more baselines especially those experience replay, some model-based methods as well as exploration method.**
>
> A: We argued in our paper that our prioritization weights can be applied to all kinds of value-based RL algorithms with Q learning. We choose DQN as baseline to show that our algorithms can boost the performance of one of the most widely studied and employed RL algorithm. Most DQN-based methods are orthogonal to our algorithm and adding prioritization weights should also improve their performance. We will test our algorithm on double DQN, as well as some model-based and offline methods.

---

> > ### Comment · Area_Chair_8DSN · 2021-08-19
> > **"A relaxation"**
> >
> > Can the authors please clarify further on "a relaxation" in Theorem 1? This is an issue that also persists in the "formal version" in the appendix. The theorem statement of Theorem 1 is vacuous when taken at face value (as it should be for any theoretical result), since any trivial upper bound of the original objective function can be claimed as "a relaxation". For the relaxation to be meaningful, one needs to quantify the difference between the relaxation and the original objective, and/or show that they use a composition of relaxation techniques that are considered tight and standard in the literature (e.g., relaxing 0/1-loss to hinge). We did not find such discussions in the paper; can the authors please comment on this issue?

---

> > > ### Author Response · Authors · 2021-08-20
> > > **Authors' Response**
> > >
> > > The relaxation includes two steps.
> > > $\eta(\pi^*)-\eta(\pi_k)\to \mathbb E_{d^{\pi_k, \pi^*}}[|Q_k-Q^*|]\to -\log\mathbb E\_{d^{\pi_k, \pi^*}}[\exp(-|Q_k-Q^*|)]$.
> > >
> > > Step 1: We use $\mathbb E_{a\sim \pi^*}Q_k(s,a)-\mathbb{E}_{a\sim \pi_k}Q_k(s,a)\leq 0$ and triangle inequality.
> > >
> > > Step 2: We apply Jensen inequality to $\mathbb E_{d^{\pi_k, \pi^*}}[|Q_k-Q^*|]$.
> > >
> > > $\mathbb E_{a\sim \pi^*}Q_k(s,a)-\mathbb{E}_{a\sim \pi_k}Q_k(s,a)\leq 0$ is not a loose bound and it is widely used in RL papers that consider regret, e.g., [Dong et al., 2019][Ayoub et al., 2020]. The objective after triangle inequality and Jensen inequality has the same minimum with respect to the original objective (here the minima is given by $Q_k=Q^*$).
> > >
> > > Approximations are necessary for solving the optimization problem in Section 3.2 and proposing practical algorithms. It is extremely difficult, if not impossible, to deal with all complex RL problems with various kinds of environments, dynamics and rewards by solving one optimization problem.
> > >
> > > **Reference**
> > >
> > > [Dong et al., 2019] Dong, S., Van Roy, B., and Zhou, Z. (2019). Provably efficient reinforcement learning with aggregated states. arXiv preprint arXiv:1912.06366.
> > >
> > > [Ayoub et al., 2020] Ayoub, A., Jia, Z., Szepesvari, C., Wang, M., and Yang, L. (2020). Model-based reinforcement learning with value-targeted regression. In International Conference on Machine Learning, pages 463–474. PMLR.

---

> > ### Comment · Reviewer_Giki · 2021-08-29
> > **thanks for clarification**
> >
> > Thanks for the detailed explanation.
> >
> > Your understanding regarding "dependence on w_k" is correct. That's exactly what I mean and your improvement makes sense.
> >
> > I still do get your explanation in 2-pi_k though. It sounds in the example you gave, then this probability will get close to 1, which is too high. Perhaps you could discuss it in the paper.

---

> > > ### Author Response · Authors · 2021-08-30
> > > **Authors' Response**
> > >
> > > Thank you for your reply. You may mean you "do not get" our explanation in $2-\pi_k$? This term is indeed close to 1 in this example but the overall sampling probability is related to a product of four terms. The other terms can have different scales. Also, $p_k$ is only proportional to the product, not equal to. Additional normalization is needed to compute the exact probability. We will give a clearer explanation in our paper.

---

### Official Review · Reviewer_8h92 · 2021-07-20

**Rating:** 8
**Confidence:** 4

**Summary:**

This paper proposes a method to improve the data selection strategy of experience replay, which is one of the standard techniques in deep reinforcement learning. As opposed to the previous method (PER), the authors provide a theoretical study to design an optimal prioritization weight by minimizing the policy regret. Some theoretical parts are partially based on the previous study (DisCor). The major contribution is that the prioritization weight is decomposed into four components. The most difficult part is to estimate $|Q_k – Q^*|$. The authors propose two practical algorithms. ReMERN estimates the upper bound of $|Q_k – Q^*| while ReMERT exploits the distance to the terminal state. The proposed method is evaluated on several benchmark tasks, including MuJoCo, MetaWorld, and Atari, and the experimental results show the proposed methods improve sample efficiency in many cases.

**Ethical Concerns:**

There are no ethical concerns.

**Limitations And Societal Impact:**

In Section 5, the authors claimed that ReMERN is robust to the randomness of environments. However, Figure 6 shows that ReMERT is more robust than ReMERN in Walker2d and Ant with noise. So, it is not clear why the two proposed algorithms are suitable for different kinds of MDP. It would be better to discuss this point in more detail.

**Main Review:**

Major comments
1. In Section 3.4.1, the authors mention that the terminal state consists of the reward only. It is correct, but some tasks do not have a terminal state. Is it possible to apply ReMERT to the non-episodic or continuing task? In fact, the performance of the proposed method is almost the same as DisCor in the HalfCheetah task.

2. I do not fully understand the TCE algorithm. For example, how do you compute $\mathbb{E}[|Q_{k-1} - \mathcal{B}^* Q_{k-2}|]$ in practice? In the simplified version (11), do you compute the expectation? In addition, it would be better to explain the difference between c of Eq.(8) and c of Eq.(9). In (8), $c$ is given by $\max_{s, a} (Q^* (s, a^*) – Q^*(s, a))$, but $c$ of Eq.(9) is the hyperparameter.

3. The authors use Likelihood-Free Importance Weights (LFIW) to estimate the density ratio $d^{\pi_k}(s, a)/\mu(s, a)$. However, I do not fully understand LFIW can estimate it in practice. LFIW assumes that the slow replay buffer contains samples from $\mu(s, a)$ while the fast replay buffer contains samples from $d^{\pi_k}$. Does the assumption hold in practice?

Minor comment

4. In Appendix C, the authors show the pseudo-codes of ReMERN and ReMERT that are integrated with DQN. Would you show the pseudo-codes for the continuous version? In particular, SAC maintains two state-action value functions to avoid the overestimation problem. Did you use the same weight for the loss functions? It would be better to show the update rules to clarify the algorithm.


**Time Spent Reviewing:**

8 hours

---

> ### Author Response · Authors · 2021-08-10
> **Authors' Response**
>
> Thank you for your constructive comments. We provide discussions and explanations about your concerns as follows.
>
>
> **Q: In section 3.4.1, is it possible to apply ReMERT to the non-episodic or continuing task?**
>
> A: ReMERT might not provide proper prioritization weights in non-episodic or continuing tasks since the "Distance to end" or $h(s,a)$ (see Section 3.4.1) can be ill-defined. In such tasks, ReMERN is a better choice.
>
>
> **Q: In the TCE algorithm, how to compute $\mathbb E[|Q_{k-1}-\mathcal{B}^*Q_{k-2}|]$? How to compute the expectation in Eq. (11)? What is the difference of $c$ in Eq. (8) and Eq. (9)?**
>
> A: We handle this term in the same way as ReMERN and DisCor algorithm by bounding it with the lower bound to $w_k$ so that $w_k$ may down-weight some transitions but never up-weight a transition by mistake. The lower bound is $c_1=\min_{s,a}|Q_{k-1}-\mathcal{B}^*Q_{k-2}|$.
>
> The expectation is about $h^{\pi_{k-1}}_\tau(s,a)$ with respect to $\tau$. In practice, the steps before reaching the terminal state is recorded once taking action $a$ from the initial state $s$. The number of steps $h(s,a)$ is simultaneously stored with the transition tuple $(s,a,r(s,a),s')$. When a certain transition tuple is sampled in Q learning, $h(s,a)$ is used to compute the Monte Carlo estimation of the expectation.
>
>
>
> $c$ is also $\max_{s,a}(Q^*(s,a^*)-Q^*(s,a))$ in Eq.(9). However, the exact value of $\max_{s,a}(Q^*(s,a^*)-Q^*(s,a))$ is not available. We regard $c$ as a hyperparameter in practice.
>
>
> **Q: Does the assumption that the slow replay buffer contains samples from $\mu(s,a)$ while the fast replay buffer contains samples from $d^{\pi_k}(s,a)$ hold in practice?**
>
> A: The fast buffer is a small FIFO queue containing data from the most recent policies. Due to limited on-policy samples, the fast buffer can not be filled with data from exactly the on-policy distribution $d^{\pi_k}$.
> Nonetheless, the data distribution in the fast buffer is much closer to $d^{\pi_k}$ than that in the slow buffer, providing adequate information to discriminate on-policy data from off-policy ones. Therefore, LFIW can still provide reasonable importance weights.
>
>
>
> **Q: Would you show the pseudo-codes for the continuous version? SAC maintains two state-action value functions to avoid the overestimation problem. Did you use the same weight for the loss functions?**
>
> A: The pseudo-codes of the continuous version is similar to that of DQN, since our algorithm applies to both continuous and discrete action spaces. As to the double value network in SAC, two error networks are trained to predict $\Delta_k$ for each of the value networks in ReMERN, while in ReMERT the loss functions share the same weights. We will add more details to the algorithm in the revised version of our paper. The source code will also be released soon for closer inspections.
>
>
> **Q: Why is ReMERT more robust than ReMERN in Walker2d and Ant with noise? Why are the two proposed algorithms suitable for different kinds of MDP?**
>
> A: The prioritization weights of ReMERT and ReMERN are both robust to the reward noise, so ReMERT has comparable performance in Walker2d and Ant with noisy reward. However, there are some other kinds of environment randomness. For example, in the Meta-World suite, the positions of objects are randomized, as clarified in Appendix D.2. In such circumstances, the "Distance to End" $h_\tau(s,a)$ in ReMERT may suffer from large variance and is hard to estimate, resulting in problematic prioritization weights. In fact, ReMERT performs slightly worse than SAC in some tasks the Meta-World benchmark, while ReMERN is more robust and can outperform SAC and DisCor in Meta-World, being more suitable for such environments with randomness. Also, in environments with limited randomness and fixed temporal ordering of states such as Ant and Humanoid, ReMERT is more suitable in that it has better sample-efficiency and time-efficiency than ReMERN.

---

### Decision · Program_Chairs · 2021-09-28

**Decision:**

Accept (Poster)

**Comment:**

The paper proposes a new method for prioritized experience replay by casting the problem of finding optimal prioritization weights as an optimization problem and then approximating its solution. The review team has significant divergence in their opinions about the paper. Multiple reviewers find the optimization-based formulation of prioritized experience replay refreshing and insightful, and are willing to accept the paper based on its novel perspective. However, as reviewers okGN and hPUt have pointed out, the theoretical derivation of the paper has the caveat that the original objective cannot be directly solved, and a fair number of relaxations and approximations need to be employed. No discussion about how those approximations affect the algorithm's ability to optimize the original objective (e.g., can you somehow quantify the approximations involved? consider questions like: do you get an additive approximation, a multiplicative one, and if so, how large are the approximation errors). While we fully acknowledge the authors' argument that it may be simply difficult to optimize the original objective and the approximations may be reasonable and necessary, this caveat still shows that the rather heuristic nature of the method (*), and in light of this, reviewer hPUt argues that the paper should be evaluated mainly on its empirical results which are not sufficiently strong (curves overlap, or the error bars are too large to tell). Given all these considerations, the AC's recommendation still leans towards acceptance, as the lack of a comprehensive theory is not a fatal issue and the rest of the paper is reasonably executed.

(*) Along this line, the AC strongly urges the authors not to call Theorem 1 a "theorem"; the statement is simply vacuous if read as a theorem, and needs to be called something less formal. The main approximation steps also need to be included for the main text to be self-contained.

**Consistency Experiment:**

NeurIPS has a long history of experimentation. In 2014, NeurIPS ran an experiment in which 10% of submissions were reviewed by two independent committees to quantify the randomness in the review process. This year, we repeated a variant of this experiment to see how the quality of the review process has changed over time.  This paper was part of the experiment and was therefore assigned to two committees (consisting of reviewers, an Area Chair, and a Senior Area Chair) that reached independent decisions.  If both committees made the same recommendation, this recommendation was followed. If a single committee recommended acceptance, the paper was accepted (with the exception of a few cases in which the other committee identified what we considered a fatal flaw, e.g., an error in a key result).

This copy’s committee reached the following decision: **Accept (Poster)**

The other committee assigned to the paper recommended **Reject**.  You can find the other set of reviews, along with any follow up discussion with the authors here:
https://openreview.net/forum?id=Ba3odanehCw